# CHANGINGGROUNDING: 3D VISUAL GROUNDING IN CHANGING SCENES

## ABSTRACT

Real-world robots localize objects from natural-language instructions while scenes around them keep changing. Yet most of the existing 3D visual grounding (3DVG) method still assumes a reconstructed and up-to-date point cloud, an assumption that forces costly re-scans and hinders deployment. We argue that 3DVG should be formulated as an active, memory-driven problem, and we introduce ChangingGrounding, the first benchmark that explicitly measures how well an agent can exploit past observations, explore only where needed, and still deliver precise 3D boxes in changing scenes. To set a strong reference point, we also propose Mem-ChangingGrounder, a zero-shot method for this task that marries cross-modal retrieval with lightweight multi-view fusion: it identifies the object type implied by the query, retrieves relevant memories to guide actions, then explores the target efficiently in the scene, falls back when previous operations are invalid, performs multi-view scanning of the target, and projects the fused evidence from multi-view scans to get accurate object bounding boxes. We evaluate different baselines on ChangingGrounding, and our Mem-ChangingGrounder achieves the highest localization accuracy while greatly reducing exploration cost. We hope this benchmark and method catalyze a shift toward practical, memory-centric 3DVG research for real-world applications.

## 1 INTRODUCTION

3D Visual Grounding (3DVG) is a critical technology that enables precise localization of target objects in 3D scenes through natural language instructions, with broad applications in service robotics (Gonzalez-Aguirre et al., 2021), computer-aided room design (Sipe & Casasent, 2003; Ganin et al., 2021), and human-machine interaction (Aggarwal, 2004; Li et al., 2020). Current methodologies and benchmarks (Achlioptas et al., 2020; Chen et al., 2020) predominantly operate under static scene assumptions, where pre-reconstructed full scene point clouds (Qi et al., 2017) and textual queries (Radford et al., 2021) are fed into end-to-end models to predict 3D bounding boxes (Jain et al., 2022; Wu et al., 2023; Luo et al., 2022; Shi et al., 2024; Guo et al., 2025) as shown in Figure 1.

However, these approaches face significant limitations when deployed in real-world robotic systems: practical environments are inherently dynamic (e.g., furniture rearrangement, object occlusion/replacement), so robots have to rescan the entire scenes to reconstruct complete point clouds every time, which is very costly(Schönberger & Frahm, 2016); otherwise, the robots don't even know whether and where the scenes have changed. In contrast, humans searching in changing environments quickly draw on memories of past scenes to pinpoint likely target areas and can complete object localization through only a few new observations. Inspired by this insight, we contend that a new memory-based paradigm for real-world 3D visual grounding is needed.

To the best of our knowledge, no existing work has explored 3D visual grounding in changing scenes by using memory from past observations. In this paper, we formally define this task and introduce a novel benchmark, the ChangingGrounding benchmark, as follows (shown in Figure 1): given the memory of the previous scene, the unexplored current scene, and a query describing the target object in the current scene, the robot needs to predict the target's 3D bounding box in the current scene. The key motivation of the task and the benchmark is to measure how a 3D visual grounding system accurately and efficiently finds the target object by leveraging the memory of past observations and exploring the current scene. So we evaluate task performance using two key metrics: the accuracy

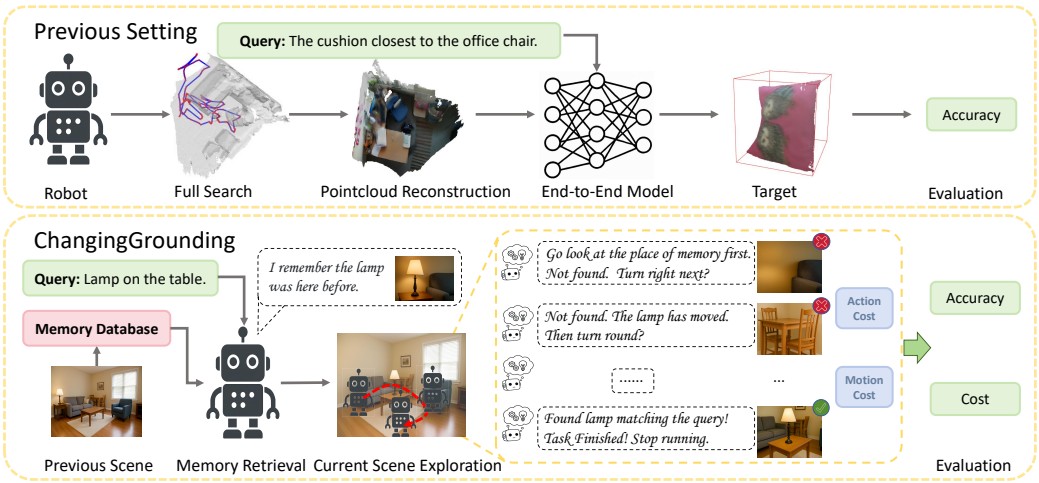

Figure 1: Comparison between the previous setting of 3DVG and the ChangingGrounding task.

of the predicted 3D bounding box and the cost for scene exploration. A better system achieves higher accuracy while keeping the lower cost. To support the task, we construct our novel dataset and benchmark, based on the 3RScan dataset (Wald et al., 2019), supported by a novel exploration and rendering pipeline to simulate how real-world robots perform 3D visual grounding.

In addition to our benchmark and dataset, we propose a novel framework called Mem-ChangingGrounder to address this new task. As current end-to-end approaches are not designed for memory access and scene agent exploration, our method is based on a zero-shot agent-based approach (Xu et al., 2024a). Specifically, Mem-ChangingGrounder first classifies user queries, then retrieves relevant memories to guide its action policy, and then explores for the target images in the scene based on this policy, next ensures fallback localization if no valid target images are found, and finally performs scanning of the target and predicts 3D localization through multi-view projection.

We introduce three additional baseline methods and compare them with our proposed Mem-ChangingGrounder on the ChangingGrounding benchmark. The three baselines simulate different grounding policies: (i) Wandering Grounding: aimless exploration, (ii) Central Rotation Grounding: simple rotation, and (iii) Memory-Only Grounding: memory-only with no exploration. Experimental results show that Mem-ChangingGrounder achieves the highest grounding accuracy among other baseline methods while maintaining a relatively low exploration cost, demonstrating a superior balance between accuracy and efficiency, and the effectiveness of our proposed policy.

## 2 RELATED WORK

**3D Visual Grounding Benchmarks and Methods.** 3D visual grounding aims to locate target objects from natural language queries. Early work focused on matching objects with shape descriptions (Achlioptas et al., 2019; Prabhudesai et al., 2020). ScanRefer (Chen et al., 2020) and ReferIt3D (Achlioptas et al., 2020) extended this to scene-level benchmarks using ScanNet (Dai et al., 2017). ScanRefer predicts full 3D bounding boxes, while ReferIt3D identifies the correct object from given candidates. Later datasets expanded the setting: Multi3DRefer (Zhang et al., 2023) supports grounding multiple objects, and ScanReason (Zhu et al., 2024a) uses complex human instructions. These benchmarks are closer to real needs but ignore temporal changes in scenes. Methods for 3D visual grounding include supervised and zero-shot approaches. Supervised models (Guo et al., 2025; Qian et al., 2024; Wu et al., 2023; Jain et al., 2022; Luo et al., 2022; Shi et al., 2024) rely on annotated datasets, combining a detection branch for 3D objects and a language branch for text encoding. They achieve strong results but are limited by scarce annotations. Zero-shot methods use LLMs (Touvron et al., 2023; Devlin et al., 2018; Brown et al., 2020; OpenAI, 2023a) and VLMs (OpenAI, 2023b; Chen et al., 2024; Liu et al., 2023; Xu et al., 2024b) to overcome this issue. Some reformulate grounding as a text problem or use LLM-generated scripts (Yang et al., 2024; Yuan et al., 2024; Fang et al., 2024). VLM-Grounder (Xu et al., 2024a) grounds objects through images instead of

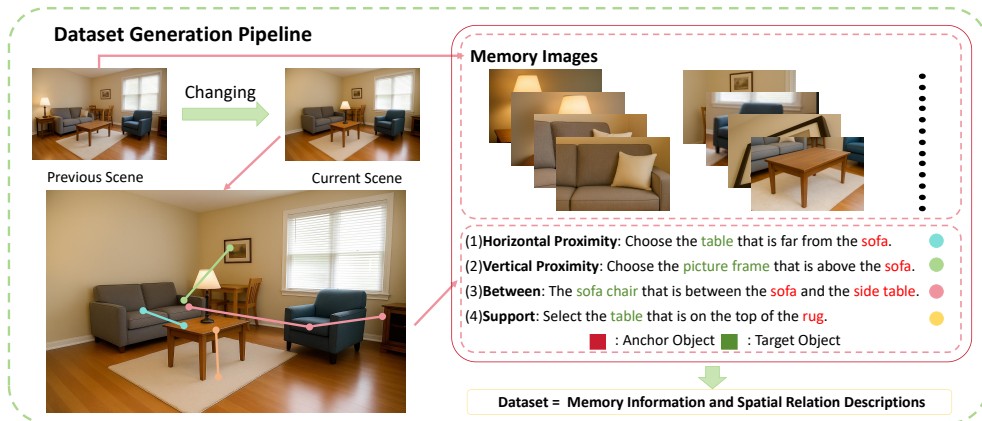

Figure 2: ChangingGrounding Dataset generation pipeline.

point clouds, and SeeGround (Li et al., 2025) selects viewpoints to render scenes for VLM input. These advances improve scalability, but none address grounding in dynamic scenes. Since VLM-Grounder does not require full point cloud input, it provides a practical basis for changing-scene grounding, and we extend our method from this framework.

**3D Perception in Changing Scenes.** Early work (Fehr et al., 2017) built a small dynamic-scene dataset for 3D reconstruction, but it lacked annotations. InteriorNet (Li et al., 2018) later provided a large synthetic dataset with object and lighting changes. 3RScan (Wald et al., 2019) pioneered the creation of a large-scale real-world indoor RGB-D dataset, encompassing scans of the same indoor environment at different time points, and introduced the task of 3D object instance relocalization, which involves relocating object instances within changing indoor scenes. Many studies followed, such as camera relocalization in changing indoor environments (Wald et al., 2020), changing detection (Adam et al., 2022), changing environment reconstruction (Zhu et al., 2024b), and changing prediction (Looper et al., 2022). Besides, Hypo3D (Mao et al., 2025) conducts a 3D VQA benchmark to evaluate models' ability in changing scenes based on 3RScan. Notably, our work represents the first exploration of 3D visual grounding tasks in changing environments. The 3RScan dataset provides scene scans at different time steps, as well as the coordinate system transformations between scenes and the correspondences of objects. We construct our novel 3D visual grounding dataset based on these annotations.

## 3 CHANGINGGROUNDING

In this section, we first formulate the ChangingGrounding task, then establish the evaluation metrics, and finally detail the dataset collection pipeline along with a statistical analysis.

### 3.1 TASK FORMULATION

Consider a robot that observed a room yesterday and acquired its scene information. When revisiting the room today, objects may have been rearranged. The robot must locate a target object described by a user query. A naive solution is to explore the whole room and then apply standard 3DVG methods, but this is inefficient. Inspired by human memory, we propose enabling robots to use past observations for more efficient and accurate grounding.

The task is defined as $\langle S_p, S_c, M_p, D_c \rangle \to B$. $B$ is the predicted 3D bounding box of the target. $M_p$ is the memory of the previous scene, including RGB-D images and poses. It is specifically defined as: $M_p = (\{I_p\}, \{P_p\})$, where $\{I_p\}$ is the set of RGB-D images from the previous scene, and $\{P_p\}$ is the corresponding set of camera poses. $S_p$ is a unified representation for all information that can be derived or extended from $M_p$ ( RGB-D + pose data ), such as reconstructed 3D point clouds. A method may freely choose whether to use $S_p$ or not: $S_p = f_{\text{scene}}(M_p)$. $D_c$ is a text description of

Table 1: Comparison of datasets. VG: Visual Grounding. CVG: Changing Visual Grounding

| Dataset | Task | Prompts Size | Build on | Dynamic Support |
|---|---|---|---|---|
| Nr3D (Achlioptas et al., 2020) | VG | 42K | Static dataset | No |
| Sr3D (Achlioptas et al., 2020) | VG | 84K | Static dataset | No |
| ScanRefer (Chen et al., 2020) | VG | 52K | Static dataset | No |
| ViGiL3D (Wang et al., 2025a) | VG | 0.35K | Static dataset | No |
| Multi3DRefer (Zhang et al., 2023) | VG | 62K | Static dataset | No |
| ScanReason (Zhu et al., 2024a) | VG | 10K | Static dataset | No |
| **ChangingGrounding (ours)** | CVG | 267K | Changing dataset | Yes |

the target object. $S_c$ is the current scene with unknown changes:

$$S_c = \begin{cases} \text{Real scene,} & \text{if agent deployed in a real environment,} \\ \text{Mesh / point-cloud,} & \text{otherwise.} \end{cases}$$

The agent needs to ground the target object in $S_c$ using $M_p$ and $D_c$ ( $S_p$ optional). In the concrete execution process, the agent needs to select actions by conditioning on the language query, the previous-scene memory, and the observations accumulated so far. Formally, at step $t$ the agent chooses $a_t = \pi(D_c, M_p, o_{1:t})$, and executing $a_t$ moves the agent to a new pose $pose_{t+1}$. At this pose, the agent obtains a new observation

$$o_{t+1} = \begin{cases} \text{CameraCapture}(S_c, pose_{t+1}), & \text{in real-world deployment,} \\ \text{Render}(S_c, pose_{t+1}), & \text{when operating on mesh/point-cloud data.} \end{cases}$$

Finally, the agent needs to integrate all information gathered throughout the process to determine the location of the target object. Because the task requires both efficient and precise grounding, we will evaluate this task by two key metrics: accuracy and exploration cost. For research simplicity, we also set several task assumptions as follows.

**Zero-cost Memory Access.** The memory information $M_p$ for the previous scene $S_p$ is stored in the robot's database and can be accessed at any time without incurring additional cost.

**Standardized Scene Coordinate System.** Each 3D scene has a standardized coordinate system $T_s$. For different temporal scene states of the same physical space, their standardized coordinate systems are aligned to one global coordinate system.

**Robot's Initial Pose.** We adopt the OpenCV right-handed camera coordinate convention and apply it to all poses. For convenience, we assume that in each scene, the robot is initially positioned at the origin of $T_s$ and its initial orientation is obtained by transforming $T_s$ so that the axes satisfy the OpenCV convention

**Exploration.** For the new scene $S_c$, the robot needs to explore to obtain relevant information about the scene. Therefore, the acquisition of information about $S_c$ will involve certain costs. The cost includes action cost $C_a$ and motion cost $C_m$ (details in Section 3.2).

**New Observations.** We assume the robot is equipped with an RGB-D camera, and it can move to achieve new positions and orientations (new poses). At the new pose, the robot can obtain a new observation. To fulfill this assumption, we developed a rendering module. The rendering module takes the mesh file of a scene and the desired new pose as inputs and outputs the RGB-D image observed from the new pose within the scene (an RGB-D image formulated as $(\mathbf{I}, \mathbf{D}) = \text{Rendering}(\text{Mesh}, Pose)$).

## 3.2 EVALUATION METRICS

The evaluation uses two metrics: localization accuracy and exploration cost. Localization accuracy follows standard 3DVG evaluation and is measured by the ratio of samples whose predicted 3D bounding box overlaps the ground-truth box above a threshold (e.g., Acc@0.25).

The exploration cost includes action cost $C_a$ and motion cost $C_m$. $C_a$ counts the number of actions taken until the target is localized. Each action means the robot moves to a new pose and captures a new observation. Action cost alone may be insufficient, since a single action can involve a large movement. We therefore also measure motion cost.

Motion cost considers both translation and rotation. To compare them on the same scale, we convert to time using nominal speeds: translation $v = 0.5\,\text{m/s}$, rotation $\omega = 1\,\text{rad/s}$. Given poses $\{(t_1, R_1), \ldots, (t_n, R_n)\}$, with $n = C_a$, the costs are: $C_{\text{trans}} = \frac{1}{v} \sum_{i=1}^{n-1} \|t_{i+1} - t_i\|$, $C_{\text{rot}} = \frac{1}{\omega} \sum_{i=1}^{n-1} \arccos\left(\frac{\text{Tr}(R_i^\top R_{i+1})-1}{2}\right)$, $C_m = C_{\text{trans}} + C_{\text{rot}}$.

It's noted that when calculating $C_{trans}$, we only consider cost on the horizontal plane. The rotation term uses the well-known trace formula $\theta = \arccos\left((\text{Tr}(R^\top) - 1)/2\right)$, which gives the rotation angle $\theta$ of a rotation matrix. By summing these angles and dividing by the nominal rotational speed $\omega$, we obtain the rotation time.

### 3.3 Dataset and Benchmark Construction

We constructed the ChangingGrounding dataset to support the proposed task. It contains: (1) spatial relation descriptions of target objects as user queries; (2) original RGB-D images of each scene with camera poses as memory information; (3) a mesh file for generating new observations. We base our dataset on 3RScan, which has 1,482 snapshots from 478 indoor environments, providing transformations between scans for alignment, dense instance-level annotations, and object correspondences across scans. These properties allow us to align scenes, re-render them, and construct cases where objects are moved.

The dataset is built in two steps. As shown in Figure 2, first, we generate spatial relation descriptions following ReferIt3D (Achlioptas et al., 2020). Second, we process 3RScan data to obtain re-rendered images and store them as memory information.

**Spatial Relation Descriptions.** We use the template ⟨Target Category⟩⟨Spatial Relation⟩⟨Anchor Category⟩, such as "the chair farthest from the cabinet." The anchor category differs from the target. We select 209 fine-grained categories from 3RScan, including those appearing in at least four scenes and those marked as rigid-move. A target is valid if it belongs to these categories and has at most six distractors of the same class. Anchor categories include these 209 classes plus 24 others. ReferIt3D defines five spatial relations (Horizontal Proximity, Vertical Proximity, Between, Allocentric, and Support), but we exclude Allocentric since 3RScan lacks front-orientation annotations. The detailed rationale for category filtering and the construction feasibility are provided in Appendix G. The set of spatial relation descriptions is provided in the supplementary material.

**3RScan Processing.** We align scans of the same scene to a global coordinate system. The initial scan is taken as the reference, and we calculate its coordinate system first. Then, transformations between the reference and other scans are applied to align all other scans to the coordinate system. For re-rendering, we adopt the ScanNet (Dai et al., 2017) camera model ($1296 \times 968$ resolution with intrinsics $(1169.6, 1167.1, 646.3, 489.9)$) and use our rendering module to standardize RGB-D images as memory frames.

**Statistics.** We compared the ChangingGrounding dataset with existing datasets in Table 1. Our ChangingGrounding is the largest and the only one built on changing environments. It introduces the new task of changing visual grounding, along with its formulation, baselines, and evaluation protocol. More details and some visual examples are presented in Appendix G and O.8.

## 4 Mem-ChangingGrounder (MCG)

In this section, we introduce Mem-ChangingGrounder (MCG), a zero-shot framework for 3D visual grounding in changing scenes. MCG takes a query $D_c$ in the current scene $S_c$ and predicts the 3D bounding box of the target object $O^t$, using memory $M_p$ of the previous scene represented as RGB-D images $\{I_p\}$ and camera poses $\{p_p\}$. As shown in Figure 3, MCG has two action policies within four core modules: Query Classification, Memory Retrieval and Grounding, Fallback, and Multi-view Projection. The workflow is to first classify the query and select the path for retrieval and grounding. MCG then explores the current scene with action policies to locate the target. If this

Figure 3: Workflow of Mem-ChangingGrounder (MCG). The upper part shows the overall pipeline: MCG classifies queries, retrieves memory, uses OSS and SRAS to search, applies fallback when needed, and predicts the 3D bounding box through multi-view projection. The lower part shows details of OSS, SRAS, and Multi-view Projection.

fails, the fallback module estimates the target. Finally, multi-view information is fused for accurate grounding. Because MCG builds on the VLM-Grounder (Xu et al., 2024a) framework, we will first introduce this framework (Section 4.1) and then present MCG's four key modules.

## 4.1 PRELIMINARY OF VLM-GROUNDER

VLM-Grounder is a zero-shot 3D visual grounding method that localizes target objects using 2D images and natural language. The pipeline is: from the current scene image sequence $\{I_c\}$, all images containing the target category are detected to form $\{I_c\}^{det}$; then a VLM (OpenAI, 2025a) analyzes the query and stitched $\{I_c\}^{det}$ to find the target image; next, an open-vocabulary detector (Liu et al., 2024) proposes objects in the image, and the VLM selects the correct one; finally, a multi-view projection module fuses multiple viewpoints to estimate the 3D bounding box.

## 4.2 ACTION POLICY

Before presenting the core modules of MCG, we briefly describe two action policies frequently employed in MCG to explore the new scene and find the target object, which are the Omnidirectional Scene Scanner (OSS) and the Spatial Relation Aware Scanner (SRAS). We give the basic explanation here, while the complete derivation is in Appendix H.

**Omnidirectional Scene Scanner.** The OSS module is a set of robot agent actions when it needs to locate an anchor object or a target object. As shown in the bottom leftmost figure of Figure 3, from a given pose, the agent performs a 360° scan by rotating around the gravity axis, ensuring a full exploration of the surroundings. The collected Images from all poses are then collected, indexed, dynamically stitched, and then sent to the VLM, which selects the one that best matches the query.

**Spatial Relation Aware Scanner.** The SRAS module defines agent actions when the agent needs to find the target after locating the anchor. It generates observations from the anchor pose using the spatial relation between anchor and target, and applies VLM (OpenAI, 2025a) to identify the target image. As shown in Figure 3, given anchor pose $p^a$ and query $D_c$, the agent first uses VLM to infer the relative position of target $O^t$ to anchor $O^a$. Based on this relation, the agent adjusts $p^a$, collects images, stitches them, and inputs them with $D_c$ into VLM to predict the target. New poses are generated for different spatial relation categories.

### 4.3 DETAILS OF MCG

**Query Classification.** From a memory perspective, user queries are either verifiable or unverifiable. For unverifiable queries, even if the target and anchor stay still, a target found in memory may not match in the current scene. For example, "the chair farthest from the table" may point to a different chair when a new one appears farther away in the current scene. Thus, the memory target no longer fits the query. In contrast, if the target and anchor stay static, and the target found in the memory will always match the query in the current scene, this kind of query is verifiable. For example, "the vase on the table" is verifiable. MCG uses the VLM to judge whether a query is verifiable.

**Memory Retrieval and Grounding.** As shown in Figure 3, this module is designed to obtain an initial estimate of the target grounding result by combining memory retrieval and exploration. This module locates the target image in the current scene $S_c$ by integrating memory $M_p$, user queries $D_c$, and exploration of $S_c$. In short, this module will first try to use the memory $M_p$ to locate an anchor object, and then explore the target object based on the spatial relationship between the anchor and the target. This process is carried out with the assistance of the two action policies, SRAS and OSS modules, which give action according to current observations and the spatial relations. Depending on the type of query, the specific grounding procedures differ. This module is carefully designed with a not simple, yet effective logic. We provide detailed explanations in Appendix J.

**Fallback.** The Fallback module will be activated to find an alternative target image if the Memory Retrieval and Grounding module fails to ground an initial estimate of the target object and return the corresponding target image. Specifically, the agent will first retrieve from memory the clearest image that contains an object of the target class. It will then start from the pose of the image and use OSS to perform a 360° search for images containing the target object as an alternative result for the Memory Retrieval and Grounding module.

**Multi-view Projection.** After memory-guided retrieval or fallback identifies the target image, the agent uses VLM (OpenAI, 2025a) to predict its 2D bounding box. The image and box are then fed to SAM (Kirillov et al., 2023) to obtain a mask, which is projected into 3D using depth and intrinsics to form a reference point cloud. Since this single-view cloud is incomplete, the module refines grounding through multi-view target-centered scanning. As shown in Figure 3, the agent circles the center of the reference cloud, collects multi-view observations, selects a 2d bounding box, projects them into 3D, clusters the clouds, and outputs a refined 3D bounding box. The complete calculation procedure is provided in Appendix K.

## 5 EXPERIMENTAL RESULTS

### 5.1 EXPERIMENTAL SETTINGS

**Dataset.** Following VLM-Grounder (Xu et al., 2024a) and LLM-Grounder (Yang et al., 2024), we randomly extract 250 validation samples for evaluation. Each sample can be categorized during evaluation based on its user query, falling into one of two types: "Unique" (only one instance of the target class in the scene) or "Multiple" (with distractors of the same class in the scene). The detailed sampling procedure is provided in Appendix O.1.

**Baselines and Implementation.** We evaluate three additional baselines, covering two scenarios: (i) using only exploration without memory. (ii) using only memory without exploration. The three baselines are organized as follows: 1). Wandering Grounding: the original VLM-Grounder(Xu et al., 2024a) approach utilizing all images and poses of scene $S_c$ from 3RScan; 2). Central Rotation Grounding: the VLM-Grounder utilizing images captured through a similar methodology of OSS at the initial pose of $S_c$; 3). Memory-Only Grounding: the VLM-Grounder utilizing images only from the memory $M_p$ in scene $S_p$. For experiments, we use GPT-4.1-2025-04-14 (OpenAI, 2025a) as the VLM, with tests in both high-resolution and low-resolution image modes. We set Temperature = 0.1, Top-P = 0.3, max stitched images $L = 6$, and ensemble images $N = 7$. The retry limit is set to $M = 3$ for baselines, but removed in MCG since it employs a different fallback. The 2D detectors include SAM-Huge (Kirillov et al., 2023) and GroundingDINO (Liu et al., 2024).

**Evaluation Metrics.** Accuracy is measured by Acc@0.25 and Acc@0.5, the percentage of samples where the IoU between prediction and ground truth exceeds 0.25 or 0.50. Cost is measured by $C_a$

Table 2: Accuracy and exploration cost of three baselines and Mem-ChangingGrounder (ours) on the ChangingGrounding benchmark under high- and low-resolution settings. The two resolution settings are separated by a middle line. Higher accuracy and lower cost indicate better performance. Best accuracy and lowest cost are bolded. Cost is measured in 1K seconds one unit.

| Method | Model | Res | Overall | | Unique | | Multiple | | Cost ↓ | | | |
|---|---|---|---|---|---|---|---|---|---|---|---|---|
| | | | @0.25 | @0.50 | @0.25 | @0.50 | @0.25 | @0.50 | $C_a$ | $C_{trans}$ | $C_{rot}$ | $C_m$ |
| Wandering Grounding | GPT-4.1 | low | 24.80 | 10.80 | **30.67** | 10.67 | 16.00 | 11.00 | 44.23 | 8.73 | 8.78 | 17.51 |
| Central Rotation Grounding | GPT-4.1 | low | 16.80 | 6.00 | 19.33 | 9.33 | 13.00 | 1.00 | 18.00 | 0.00 | 1.70 | 1.70 |
| Memory-Only Grounding | GPT-4.1 | low | 20.80 | 10.00 | 22.67 | 10.67 | 18.00 | 9.00 | **0.00** | **0.00** | **0.00** | **0.00** |
| Mem-ChangingGrounder (ours) | GPT-4.1 | low | **29.20** | **14.80** | 30.00 | **15.33** | **28.00** | **14.00** | 8.53 | 5.73 | 3.98 | 9.70 |
| Wandering Grounding | GPT-4.1 | high | 32.40 | 12.80 | 38.67 | 16.00 | 23.00 | 8.00 | 44.23 | 8.73 | 8.78 | 17.51 |
| Central Rotation Grounding | GPT-4.1 | high | 17.20 | 6.80 | 18.00 | 8.00 | 16.00 | 5.00 | 18.00 | 0.00 | 1.70 | 1.70 |
| Memory-Only Grounding | GPT-4.1 | high | 26.00 | 12.40 | 26.67 | 11.33 | 25.00 | 14.00 | **0.00** | **0.00** | **0.00** | **0.00** |
| Mem-ChangingGrounder (ours) | GPT-4.1 | high | **36.80** | **18.00** | **42.67** | **19.33** | **28.00** | **16.00** | 8.47 | 5.84 | 3.92 | 9.76 |

Table 3: Memory strategy.

| Memory | Acc. | $C_a$ | $C_m$ |
|---|---|---|---|
| w/o. | 35.2 | 31.94 | 18.60 |
| w. | 36.8 | 8.47 | 9.76 |

Table 4: Fallback.

| Fallback | Acc. | $C_a$ | $C_m$ |
|---|---|---|---|
| w/o. | 36.4 | 8.21 | 9.53 |
| w. | 36.8 | 8.47 | 9.76 |

Table 5: Multi-view projection.

| Strategy | Acc. | $C_a$ | $C_m$ |
|---|---|---|---|
| Baseline | 22.4 | 4.81 | 2.95 |
| +Multi-scan | 28.0 | 8.52 | 9.72 |
| +filter | 36.8 | 8.47 | 9.76 |

and $C_m$ (defined in Section 3.2). Details how $C_a$ and $C_m$ are computed for each baseline methods and MCG are in Appendix M.

## 5.2 MAIN RESULTS

As shown in Table 2, our Mem-ChangingGrounder (MCG) achieves the best accuracy in both low- and high-resolution settings (29.2% and 36.8%), outperforming all baselines. That clear margin underlines the superiority and robustness of our solution for grounding performance across a spectrum of visual qualities. At the same time, our method maintains modest action cost $C_a$ and motion cost $C_m$, which demonstrates a carefully engineered compromise between effectiveness and efficiency. This is because MCG consults memory before moving and then performs short, targeted actions, avoiding long exploratory loops.

**Wandering Grounding (WG):** In comparison, the WG method achieves the second-highest accuracy at both resolutions, but its $C_a$ is about five times larger and its $C_m$ is also much higher. The reason is its wide roaming: the robot repeatedly sweeps across the environment. This traversal lets the agent collect more scene information and improve accuracy, but also forces long travel and many actions, which cause a heavy cost.

**Central Rotation Grounding (CRG):** The CRG method keeps the robot at the scene center and performs one full rotation, which removes translation and reduces actions, making the cost very low. However, this single and constrained viewpoint misses occluded objects, height-shifted objects, or complex spatial layouts, so important visual information is lost. As a result, its grounding accuracy is the lowest among all methods.

**Memory-Only Grounding (MOG):** The MOG method also has a low cost since it relies only on stored panoramic memories, with one final adjustment after estimating the target. If the memories are complete and the scene unchanged, accuracy can be high. But if the environment changes or the memory has gaps, the lack of verification and correction quickly reduces accuracy, placing this method far behind ours.

Table 6: VLMs capacity.

| Method | Acc. | $C_a$ | $C_m$ |
|---|---|---|---|
| GPT-4o | 31.6 | 8.34 | 8.47 |
| GPT-4.1. | 36.8 | 9.52 | 9.76 |

Table 7: Render vs real.

| Method | Acc. | $C_a$ | $C_m$ |
|---|---|---|---|
| w. | 28 | 1.74 | 2.05 |
| w/o. | 24 | 1.62 | 1.94 |

Table 8: 3DVG comparision.

| Method | Acc. | $C_a$ | $C_m$ | Inf(s) | T(s) |
|---|---|---|---|---|---|
| 3D-Vista | 33.2 | 18.00 | 2.39 | 0.05 | 12.73 |
| MCG | 36.8 | 8.47 | 9.76 | 113.2 | 152.24 |

Table 9: Module success rates (%). Sub-module accuracies of MRGS and MS are separated by bars.

| **Total** | QAS | QCS | MRGS | MS | **MRGS** | VP | VCI | **MS** | OD | SRI | MP |
|---|---|---|---|---|---|---|---|---|---|---|---|
| 36 | 100 | 100 | 56 | 64 | 56 | 70 | 80 | 64 | 89 | 96 | 75 |

Table 10: Inference time (s) of different modules.

| **Total** | View Preselection | VLM Predict Images | Detection Predictor | Project Points |
|---|---|---|---|---|
| 113.2 | 11.3 | 11.7 | 0.2 | 0.7 |

Table 11: Different Prompts.

| Prompts | Acc. | $C_a$ | $C_m$ |
|---|---|---|---|
| v_origin | 42 | 1.60 | 2.05 |
| v_less | 40 | 1.61 | 2.02 |
| v_fix_layout | 36 | 1.20 | 1.45 |

Table 12: Different VLMs.

| Model | Acc. | $C_a$ | $C_m$ |
|---|---|---|---|
| gpt | 42 | 1.60 | 2.05 |
| gemini | 38 | 1.60 | 1.83 |
| claude | 24 | 1.68 | 2.17 |

Table 13: Different uncertainty.

| Temp. & Top_p | Acc. | $C_a$ | $C_m$ |
|---|---|---|---|
| 0.1, 0.3 (origin) | 42 | 1.60 | 2.05 |
| 0.5, 1.0 | 44 | 1.74 | 2.10 |
| 0.7, 1.0 | 42 | 1.62 | 2.02 |

Overall, our method reaches the highest accuracy while keeping $C_a$ and $C_m$ low, proving that memory-augmented strategies balance efficiency and precision in changing scenes. Memory-Only Grounding and Central Rotation Grounding cut costs but lose accuracy by avoiding exploration or using oversimplified strategies. Wandering Grounding explores more but ignores memory, so it needs many actions and long travel, leading to higher costs and lower accuracy than ours.

## 5.3 ABLATION STUDIES

We validated several design choices in MCG. For the **memory strategy**, we compared with a memory-free setting where the system follows Wandering Grounding's pose sequence without memory. As shown in Table 3, both achieve similar accuracy, but the memory-free approach consumes far more costs, confirming the efficiency of memory use. For **fallback**, we tested the method without this strategy. As shown in Table 4, though accuracy and cost are similar with or without fallback, adding fallback ensures completeness and coverage of edge cases. For **multi-view projection**, we performed a two-step ablation: first, adding center rotation for multi-view acquisition, then adding outlier removal. As shown in Table 5, each step improved accuracy; although center rotation increases cost, it benefits the localization accuracy. For application scenarios where an accurate 3D bounding box is not necessary, the cost of MCG could be further reduced. Finally, for **different VLMs**, we compared GPT-4o (OpenAI, 2024) and GPT-4.1 (OpenAI, 2025a). As shown in Table 6, costs are similar, but GPT-4.1 yields higher accuracy, indicating that better VLM directly results in better performance. Also, we compare rendered versus real memory images and find that rendering doesn't have a significant negative impact on grounding accuracy, as shown in Table 7. More detailed analysis of the rendered-versus-real experiment is in Appendix O.2.

## 5.4 DISCUSSION ABOUT CRITICAL LIMITATIONS OF 3D VISUAL GROUNDING METHODS

Although existing 3D visual grounding methods may rescan the entire scene each time to perform our changing grounding task, the approach is still impractical. In dynamic environments,scenes change frequently, it is unrealistic to perform a full rescan every time an object is moved. Worse yet, the system often does not know whether, when, or where the scene has changed, making it difficult to decide whether a new scan is necessary before grounding. This reliance on complete

and repeated reconstruction is inefficient and infeasible in practical applications. Nevertheless, for a more complete comparison, we adapted the 3D-Vista (Zhu et al., 2023) model to the memory-based setting; it is pre-trained on 3RScan and has learned the SR3D text distribution. It should be noted that 3D-Vista requires ground-truth bounding boxes, which makes its performance higher than realistic. We also use a simplified way to calculate cost, which makes the cost lower than it is. As shown in Table 8, our method still outperforms it regarding accuracy and cost.

We also provide a time comparison between the 3D-Vista method and MCG. Besides inference time, we include a comparison of the total processing time. This is because traditional 3D methods take point clouds as input, and under our setting, the time from acquiring images to reconstructing the point cloud must also be considered. For 3D-Vista, the total time consists of three parts: the time to acquire RGB-D images (this part of the time is given by the average exploration cost $C_m$), the 3D reconstruction time (we approximate the cost using the time an advanced 3D reconstruction method ( e.g., VGGT (Wang et al., 2025b) ) takes to process 100 images) , and the inference time. For MCG, we need to account for the average exploration cost $C_m$ and its inference time.

We acknowledge that the inference speed still has significant room for improvement, as this is a research project. As shown in Table 17, most of the time in our pipeline is spent on VLM inference. With the rapid progress of VLM technology, we expect that high-speed large models will soon be available on edge devices. For example, the FastVLM project introduced by Apple (Vasu et al., 2025) achieves an 85× faster TTFT compared with LLaVA-OneVision at 1152×1152 resolution. This progress opens promising opportunities for greatly reducing the overall runtime of our method.

### 5.5 DISCUSSION REGARDING VLMs

To further investigate how VLMs affect the experimental results, we examine three categories of VLM-related factors: different VLM prompt designs, different underlying VLMs, and different levels of VLM uncertainty. As shown in Table 11, Table 12 and Table 13, simplified prompts and higher-temperature settings showed minimal impact, while fixed-layout prompting and certain VLMs (e.g., claude (Anthropic, 2025)) led to more noticeable degradation. Detailed analyses are provided in Appendix O.4.

### 5.6 SUCCESS RATE AND INFERENCE TIME

We randomly sampled 50 examples from the test samples and checked the success rate of important stages of our pipeline. The following defines the criteria for whether a step is successful. (1) Query Analysis Stage (QAS): correctly extracts both the target category and any additional constraints.( This accuracy component is related to the preprocessing of the test data, which is similar to VLM-Grounder (Xu et al., 2024a) ). (2) Query Classification Stage (QCS): assigns the query to the proper categories. (3) Memory Retrieval and Grounding Stage (MRGS): picks a view that contains the target object. (4) Multi-view Stage (MS): The 3D IoU between the predicted box and the ground-truth box is $\geq 0.25$. Specifically, the success rate in the MRGS depends on 2 other modules: (a) View Pre-selection (VP) and (b) VLM Choose Image (VCI). The MS depends on 3 other modules: (a) OV Detection (OD); (b) Select Reference Instance (SRI); (c) Multi-view Projection (MP). These modules' detailed explanations are in Appendix O.6. As shown in Table 9, the largest sources of error stem from the MRGS and the MS. Detailed failure cases and descriptions are provided in Appendix O.7 and Appendix N. Also, we report the inference time of different modules in Table 10, with more analysis in Appendix O.6.

## 6 CONCLUSION

In this work, we reformulate 3D visual grounding as an active, memory-driven task and introduce ChangingGrounding, the first benchmark for changing scenes with cost accounting. We also propose a novel and strong baseline named Mem-ChangingGrounder for this new task. Mem-ChangingGrounder demonstrates that leveraging memory and efficient exploration can raise localization accuracy while cutting down grounding costs. We believe our dataset, task, and baselines will motivate and serve as a starting point for future research on 3D visual grounding in changing scenes. More details (e.g., use of LLMs, open problems, etc.) are in the appendix.

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

## A  APPENDIX OVERVIEW AND ORGANIZATION

This appendix provides supplementary details to support and extend the main paper. The organization of the appendix is as follows:

1. **Use of LLMs ( Section B ):** This section states the use of Large Language Models (LLMs).

2. **Ethics Statement ( Section C ):** This section provides ethics statements.

3. **Reproducibility Statement ( Section D ):** This section provides reproducibility statements.

4. **Broader Impact ( Section E )** The broader societal implications of our work are discussed.

5. **Benchmark Statement ( Section F ):** This section describes the release status and usage policy of the ChangingGrounding Benchmark (CGB).

6. **More Details for Data Construction ( Section G ):** This section describes more details for the data construction process of CGB.

7. **Action Policy ( Section H ):** This section shows a more detailed description of action policies, the Omnidirectional Scene Scanner (OSS), and the Spatial Relation Aware Scanner (SRAS)

8. **Query Classification ( Section I ):** This section presents additional details of the Query Classification module.

9. **Memory Retrieval and Grounding( Section J):** This section presents a more detailed explanation of two different algorithmic paths based on the query classification results.

10. **Multi-view Projection ( Section K ):** This section introduces more details for the Multi-view Projection module. This module obtains multi-view point clouds and then filters them to get more accurate 3D bounding boxes.

11. **VLM Prompts ( Section L ):** We provide the full list of vision-language prompts used in MCG, covering all modules including memory retrieval, spatial relation parsing, multi-view comparison and selection, and fallback strategies. These prompts form a modular, interpretable interface for multi-stage reasoning.

12. **Cost Calculation for Methods ( Section M ):** This section details how action costs and motion costs are computed for each method. The evaluation aligns with the cost metrics defined in the main text, and a note explains that all costs are reported in units of 1,000 seconds (e.g., 9k = 9000s).

13. **Open Problems ( Section N ):** We outline the current limitations of the CGB benchmark and the MCG method, including the lack of allocentric relations, the impact of rendering noise, and the dependency on external 2D models. Future improvements are discussed.

14. **More Results ( Section O ):** Additional results are presented to assess the robustness of MCG, including detailed sampling procedure, a comparison between using rendered vs. real images in memory, human accuracy, a detailed discussion regarding VLMs, and a set of failure cases analyzing the limitations of VLM, SRAS, SAM, and the projection pipeline. A complete example is shown to illustrate how MCG grounds a target object in a changing scene.

## B  USE OF LLMS

We used large language models (OpenAI's ChatGPT (OpenAI, 2024)) solely to aid in the polishing of English writing. The models were employed to improve clarity, grammar, and style in the manuscript text. No part of the research design, data analysis, model implementation, or results interpretation was generated or influenced by LLMs. All scientific contributions, ideas, and experimental results are entirely the work of the authors.

## C    ETHICS STATEMENT

This work focuses on the design and evaluation of a benchmark. It does not involve human subjects, sensitive personal data, or private information. All datasets used are publicly available. We adhere to academic integrity standards.

## D    REPRODUCIBILITY STATEMENT

To lower the research threshold and enable independent fairness audits, we will fully open-source our benchmark generation process, data preprocessing methods, and evaluation scripts. All data are drawn from public or simulated scenes and contain no personally identifiable information.

## E    BROADER IMPACT

This study introduces a new task for changing scene 3D visual grounding, releasing the open benchmark CGB and a strong reference method, MCG. This technology can significantly enhance the efficiency of logistics and service robots in dynamic environments, advancing smart manufacturing and supply-chain management. However, rapid automation may temporarily displace low-skill jobs, requiring joint reskilling programs to equip workers with digital skills.

## F    BENCHMARK STATEMENT

We will publicly release the proposed CGB benchmark and its accompanying dataset on the Huggingface platform, making it freely accessible to the research community. The dataset will be regularly updated and maintained to ensure its accuracy and relevance. It is important to note that, at this stage, all available data in the CGB benchmark is used exclusively for testing purposes. We hope this benchmark will encourage further research into 3D visual localization in dynamically changing environments. All files within the CGB benchmark are strictly intended for non-commercial research purposes and must not be used in any context that could potentially cause harm to society.

Also, to support reproducibility, we provide benchmark testing examples for the proposed MCG method on the GitHub platform, along with detailed environment specifications and a complete execution pipeline to facilitate efficient replication and verification of experimental results.

## G    MORE DETAILS FOR DATA CONSTRUCTION

**Detail Explanation for Building Spatial Relation Descriptions.** When selecting target and anchor categories, we followed several principles from the ReferIt3D benchmark to ensure both robustness and task relevance.

**For target categories**, we excluded classes appearing in fewer than four scenes to reduce long-tail bias from infrequent categories. In addition, we included objects that undergo changes across scenes, so the final 209 target categories are the union of objects appearing in at least four scenes and those objects exhibiting changes. This dual criterion for target category selection can reduce long-tail effects and ensure the task remains relevant to dynamic scenes.

To further maintain annotation reliability, for each individual scene, we applied the constraint that a target object must have no more than six distractors of the same category in the scene. This was motivated by ReferIt3D annotator feedback showing that error rates rise significantly when distractors exceed six. Importantly, this constraint applies per scene: a category may be excluded as a target in one scene but remain valid in another if the distractor number is within the limit.

**For anchor categories**, we followed a similar strategy, using the 209 target categories plus 24 additional large or frequent objects (e.g., fireplaces, televisions). This design improves diversity while also improving reliability, since such larger anchors are easier to describe in spatial relations. We also enforce at most one anchor object in complex scenes, because our descriptions use spatial templates (Target–Relation–Anchor) rather than detailed attributes; with multiple anchors, it would be unclear which instance is referenced. Overall, this filtering strategy balances statistical robustness

with task specificity, yielding a diverse set of over 200,000 prompts while ensuring clear and reliable grounding cases.

**Statistic.** The dataset contains 266,916 referential descriptions that uniquely locate targets through spatial relations. As shown in Figure 4, the word cloud highlights frequent terms such as common furniture, along with many less frequent items. We also merge the original 528 object categories in 3RScan into 225 broader ones for tractability, with the help of ChatGPT-o1 (OpenAI, 2025b).

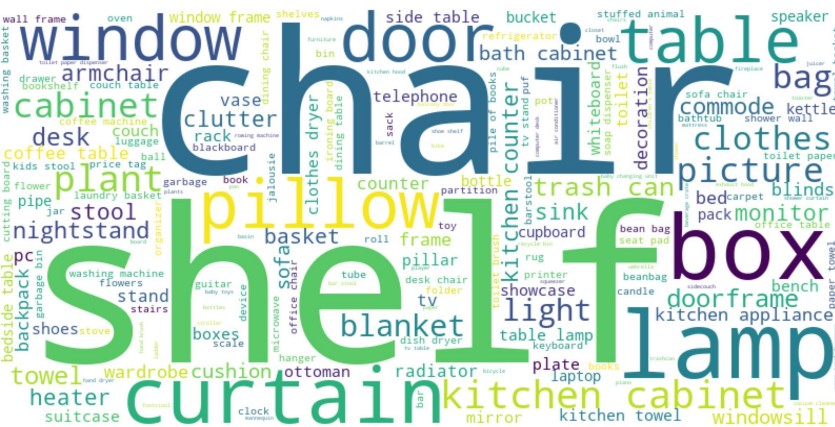

Figure 4: A word cloud generated from spatial-relation descriptions, visually highlighting the frequency of occurring terms.

## H    ACTION POLICY

**Omnidirectional Scene Scanner.** The OSS module is a set of robot agent actions when it needs to locate an anchor object or a target object. From a given pose, the agent will perform a full 360° scan and use the VLM to identify the observation that best matches the given query. As shown in the leftmost figure at the bottom of Figure 3, the agent starts from an initial pose $p$ and a user query, then generates twenty poses by rotating $p$ around the gravity axis $(\psi)$ in steps of $18° \times i$ for $i = 0, 1, \ldots, 19$. These actions ensure a comprehensive exploration of the surroundings. Subsequently, the agent tilts each pose downward by $20°$ around the horizontal $(\phi)$ axis to avoid missing objects that lie slightly below the original level. Next, the agent obtains images at each pose, annotates sequential IDs, and dynamically stitches them. Finally, the agent will input the stitched result to VLM to predict the correct image containing the anchor object or target object based on user queries.

$$\mathbf{p}_i = \mathbf{p} \cdot \mathbf{R}_\psi(18° \times i) \cdot \mathbf{R}_\phi(-20°), \quad i = 0, 1, \ldots, 19 \tag{1}$$

**Spatial Relation Aware Scanner.** The SRAS module is a set of robot agent actions when the anchor object has already been located, and its next step is to search for the target object. It is designed to obtain a series of observations starting from the anchor object pose based on the spatial relationship between the anchor and the target object, and then use VLM (OpenAI, 2025a) to predict which of these observations contain the desired target object. As shown in the second image at the bottom of Figure 3, given the anchor image pose $p^a$ and the user query $D_c$, the agent will first use VLM to analyze the positional relationship between the target object $O^t$ and the anchor object $O^a$ based on the query. Leveraging this positional relationship, the agent then adjusts $p^a$ to generate a series of new poses. Next, the agent obtains images at these new poses, assigns them unique IDs, and dynamically stitches them together. Finally, the agent inputs stitched images and $D_c$ into the VLM to predict the target image.

New poses are generated based on different categories of spatial relationships listed below:

- Horizontal and Between - The agent applies the same omnidirectional scanning strategy as the OSS module to process $p^a$ to acquire a set of new poses. Then the agent images at these poses and uses VLM to evaluate which image indeed contains the $O^t$ matching the $D_c$.

- Support and Vertical - If VLM analysis shows that $O^t$ is below $O^a$, the agent will generate a series of new poses by rotating pose $p^a$ downward around its local horizontal ($\phi$) axis in $20°$ increments. Besides tilting directly downward, in order to cover a wider exploration area, the agent will also first rotate $p^a$ a little left and right around its gravity axis ($\psi$), and then rotate downward to generate more poses. Next, the agent obtains observation images at these poses and uses VLM to evaluate which image indeed contains the $O^t$ matching the $D_c$. If $O^t$ is above $O^a$, the process is similar to that of the "below" relationship, except that the rotation is upwards.

Now on, we'll use the X, Y, and Z axes to explain more details of the pose generation methods based on different spatial relationships. This will help readers follow the accompanying code more easily. Additionally, to simplify later rotation and translation steps, we first adjust the camera pose of the anchor-object image consistently: the Y-axis points downward, and the X-axis points to the right.

**Up.** The camera is first shifted backward along its Z-axis to broaden the view. Next, we rotate the pose around its local Y-axis by $-90°$, $-45°$, $0°$, $45°$, and $90°$. For each of these turns, we add an extra tilt around the local X-axis by $0°$, $18°$, $36°$, and $54°$. This nested sweep yields $5 \times 4 = 20$ new poses, providing a more comprehensive upward field of view.

**Down.** The "down" case follows a process highly similar to the "up" case, with the key difference being the direction of rotation around the local X-axis (which controls the up-down viewing direction). The camera is first shifted backward along its Z-axis to broaden the view. Next, we rotate the pose around its local Y-axis by $-90°$, $-45°$, $0°$, $45°$, and $90°$. For each of these turns, we add an extra tilt around the local X-axis by $0°$, $-18°$, $-36°$, and $-54°$. This nested sweep yields $5 \times 4 = 20$ new poses, providing a more comprehensive downward field of view.

**Horizontal and Between.** For simplicity, we use the same procedure to generate new poses for both "horizontal" and "between" relations (Note that for the "between" relation, the initial anchor-object image only needs to include one of the two anchor objects involved in that relation). First, the camera moves backward along its local Z-axis to widen the view; next, it moves closer to the room's center to cover more of the scene; then, it rotates around its local Y-axis in $18°$ increments from $0°$ to $342°$, creating 20 evenly spaced horizontal angles; after each Y-rotation, the camera tilts $25°$ downward around its local X-axis to avoid missing lower parts of the scene. This sweep produces 20 viewpoints that give a broad, slightly downward-looking perspective of the environment.

## I  QUERY CLASSIFICATION

As stated in the main text Section 4.3, queries with "between" relation should be categorized as verifiable queries. The "between" relation is complex because it involves two anchor objects. If we followed the verifiable workflow, we would need to confirm both anchors and the target object's final position, and then we may need to build another suitable memory-retrieval and grounding algorithm based on the confirmation results. This is too complex for our current scope. For simplicity, we just mark queries with the "between" relation as unverifiable and only use the first anchor object. The remaining steps use the same procedure as the queries with a "horizontal" relation.

## J  MEMORY RETRIEVAL AND GROUNDING

Here are in-depth explanations of 2 algorithmic paths for different kinds of queries.

- - Unverifiable Queries – As mentioned in the Query Classification module, for unverifiable queries, the agent cannot ensure that the target object, which is directly grounded in memory, still matches the query in the current scene. Therefore, the agent prioritizes finding the anchor object from memory. The agent first follows the VLM-Grounder (Xu et al., 2024a) approach to preprocess images from memory: a 2D open-vocabulary detector (Liu et al., 2024) filters all images in $M_p$ to generate a preprocessed image sequence $\{I_p\}^{det}$ containing anchor class objects, which are then dynamically stitched with ID annotations. After that, the agent uses VLM to predict an image $I_p^a$ which shows the anchor object clearly from $\{I_p\}^{det}$. The agent obtains the pose $p^a$ where the image $I_p^a$ was taken, then it will go to the same pose in the current scene $S_c$ to get a new observation $I_c^a$. If the anchor object

stays still in $I_c^a$, the agent will use the spatial relationship of the query $D_c$ to find the target. Specifically, the agent inputs $D_c$ and the pose $p^a$ into the SRAS module for final target localization in $S_c$. If the anchor object doesn't stay still, the agent will go to the center of $S_c$ and directly search around to find the target. Specifically, the agent initiates OSS at the center of $S_c$ to directly locate the target.

- - Verifiable Queries – Different from unverifiable queries, for verifiable queries, the agent prioritizes directly grounding the target object matching the $D_c$ from memory. After a similar pre-process pipeline as verifiable queries, the agent obtains stitched images $\{I_p\}^{det}$ that contain the anchor class objects or target class objects. Then it uses VLM (OpenAI, 2025a) to select from $\{I_p\}^{det}$ a target image $I_p^t$ containing target object satisfying the $D_c$ and an anchor image $I_p^a$ containing anchor object. Next, by moving to the same camera poses $p^t$ and $p^a$ of $I_p^t$ and $I_p^a$ in the current scene $S_c$, the agent obtains the corresponding new observations $I_c^t$ and $I_c^a$. Following that, the agent verifies the status of images $I_c^t$ and $I_c^a$. If the target object in $I_c^t$ and the anchor object in $I_c^a$ both stay still, the agent directly outputs $I_c^t$ as a result. If the target object doesn't stay still but the anchor object stays still, the agent will use the spatial relationship of $D_c$ to find the target starting from the anchor pose $p^a$. Specifically, the agent will invoke SRAS and input $D_c$ and anchor image pose $p^a$ for localization. If the anchor object moves, the agent will first try to locate it in $S_c$, and then use the relationship to find the target. This is because, for this type of query, once the anchor is found, the target can usually be located through a series of clear actions. Specifically, the agent will move to the center of $S_c$ and use OSS for the anchor position. It should be noted that the rotational search via OSS can terminate early: as soon as the VLM spots the anchor object, the scan stops. Once the anchor is located, the agent finally invokes SRAS to track the target.

## K    MULTI-VIEW PROJECTION

Inspired by VLM-Grounder, our multi-scan projection also merges point clouds from several views to build the final cloud. But unlike VLM-Grounder, which uses PATs (Ni et al., 2023) to get appropriate views, we gather views by scanning around the target object. The entire pipeline can be divided into three stages: (1) obtaining a reference point cloud for the target object, (2) performing surround-view scanning around the reference point cloud's center to collect multi-view point clouds, and (3) removing outliers from the aggregated point cloud set. In the main text, we have already clearly described the overall pipeline of the Multi-view Projection module. Here, we first outline the more complete process and then provide the notable details for stages 1 and 2.

After memory-guided retrieval or fallback identifies the target image, the agent will use VLM (OpenAI, 2025a) to predict the 2D bounding box of the target object detected in the image. It then feeds the image with this box to SAM (Kirillov et al., 2023) to obtain a segmentation mask, projects the mask into 3D space using depth and intrinsics, and derives a reference point cloud. However, this reference point cloud is not complete. It is derived from a projection of the target object from a single viewpoint, which may not capture all parts of the object and thus results in an incomplete point cloud. To compensate for incomplete single-view point clouds, we introduce this module to refine the grounding result with a multi-view, target-centered scanning strategy. In this module, the agent circles the center of the reference 3D point cloud to get multi-view observations and projects these observations into 3D point clouds. Finally, the agent clusters and filters these point clouds and outputs a more accurate 3D bounding box.

Specifically, from the reference point cloud, the agent extracts the 3D bounding box and computes the box's center $c$ and the diagonal length $l_{\text{box}}$. The agent uses these values to define an observation sphere. The center of this observation sphere is $c$, and the radius of this sphere is calculated as $r = \max(l_{\text{box}}/2,\ 1.5\,\text{m})$. The agent then generates sixteen poses and obtains their corresponding observations on a 30°-tilted ring around the sphere. Subsequently, the agent uses VLM to select the four observations that most clearly and completely capture the target object. For each frame, the agent will select a single valid mask for the target object: it runs an open-vocabulary detector [3] to locate the object's candidate 2D bounding boxes; segments those boxes with SAM to produce candidate masks; projects the masks into the 3D point cloud; finally keeps the one mask whose corresponding point cloud centroid is closest to reference point cloud center $c$. All valid masks are

then projected to 3D point clouds. Finally, to filter the outliers, the agent sorts these clouds by the volume of their bounding boxes and discards any cloud whose volume is substantially larger than that of the next smaller one. The remaining clouds are then fused with the reference cloud to produce the refined point cloud.

**Getting the Reference Point Cloud.** To get the reference point cloud, we need to obtain the target object's 2D bounding box and use the SAM model to get its mask in the image. Next, we can project this mask into 3D space to obtain the object's reference point cloud using the camera parameters and depth data. Therefore, first, the agent feeds the image containing the target object into GroundingDINO and removes any 2D boxes that are too large, since some boxes cover the whole image. (as GroundingDINO may occasionally return boxes covering the entire image). After that, it marks the centers of the remaining boxes on the image. Then it passes the image, the user query, and additional contextual cues (e.g., "the object is located in the bottom-right corner") into the VLM to identify the most semantically relevant 2D bounding box corresponding to the target object. The agent uses this box and its center as a positive point for SAM to create a segmentation mask. Finally, the mask is projected into a 3D point cloud using the camera parameters and the depth image, with the same denoising process strategy as VLM-Grounder during projection.

**Surround-view Scanning.** The agent scans around the reference point cloud's center to capture many new views. For each view, it runs GroundingDINO to find 2D boxes. It projects each box into 3D and measures the Euclidean distance between that box's point-cloud center and the reference center. The box with the shortest distance is kept as the target object in that view. The agent repeats this for all views and gathers the resulting point clouds. It should be noted that we also apply an initial denoising step during candidate box selection in this stage, except for the outlier removal strategy based on bounding box size sorting described in the main text. The initial denoising step is explained below.

Among the bboxes, we select the one whose center is closest to that of the reference point cloud. However, due to the limitations of the 2D object detector and SAM, this nearest candidate may not always correspond to the true target object. To address this, we first input the reference image into a vision-language model (VLM) to assess whether the target object is particularly large or partially outside the camera view. If so, no additional filtering is applied. Otherwise, we enforce a spatial constraint requiring that the center of the selected candidate point cloud lies within 0.25 meters of the reference center; this helps prevent the inclusion of significant noise points unrelated to the target object.

## L  VLM Prompts

For the baseline methods, we use the same prompts as those employed in VLM-Grounder. For the MCG method, we introduce several additional prompts, including those designed for the memory retrieval image module, prompts used to compare whether the target object has moved between images, prompts used in SRAS, and prompts applied in the multi-scan projection process. We will explain each of them in the following sections.

The **memory_retrieval_prompt_for_unverifiable_queries** selects the top 3 images that clearly capture the anchor object from a video sequence when no reliable grounding information is available. In contrast, the **memory_retrieval_prompt_for_verifiable_queries** performs a two-stage reasoning process: it first searches for images that satisfy the query constraints and falls back to identifying the target object if constraints are unmet. The **oss_prompt_for_unverifiable_queries** focuses on selecting the single image that most clearly and completely depicts the target object from a 360-degree scan, while the **oss_prompt_for_verifiable_queries** incorporates a three-step reasoning strategy, identifying the earliest image containing the anchor and then limiting the search space for target localization accordingly. The **relation_parsing_prompt** is used to infer the spatial relation (e.g., up, down, near, far, between) between the target and anchor objects from the query. The **sras_choose_target_prompt** performs target selection under a 360-degree rotation by evaluating multiple views and returning the most confident match. The **compare_prompt** determines whether two images captured from the same pose show the target object at the same position, supporting consistency checks. The **fallback_prompt** implements a robust two-step procedure: locating a query-matching image if available, or falling back to the clearest image showing the object class. The **get_good_view_prompt** is used to retrieve up to four images that provide the best views of a ref-

erence object based on a reference image with a bounding box. Finally, the **bboxchoose_prompt** refines object selection by identifying the most probable target object among multiple candidate boxes, integrating query content and spatial descriptions. Together, these prompts provide a structured, interpretable, and modular interface for vision-language agents to perform complex multi-view spatial reasoning and object grounding tasks. The textbflimit_prompt guides the VLM to assess whether the target object is overly large or partially occluded, serving as a prior for filtering unreliable candidate point clouds. Table 14 shows their detailed contents.

Table 14: VLM_prompts

---

**memory_retrieval_prompt_for_unverifiable_queries**

---

You are an intelligent assistant proficient in analyzing images. Given a series of indoor room images from a video, you need to analyze these images and select the best 3 images. Each image has an ID in the upper left corner indicating its sequence in the video. Multiple images may be combined and displayed together to save the place. The anchor object is {anchor_class}. If there are some images that are very similar, only select the clearest one to participate in the further selection process. Select the best 3 images from the remaining images according to the following rule: Rule 1: Select those images from the remaining ones that can clearly display the anchor object until the total number of selected images reaches 3. Please reply in json format, including "reasoning" and "selected_image_ids":

{
"reasoning": "Your reasoning process", // Your thinking process regarding the selection task
"selected_image_ids": ["00045", "00002", "..."], // A list of the IDs of the best 3 images selected according to the rules. Note that the returned IDs should be in the form of "00045", not "00045.color", and do not add any suffix after the numbers.
"unique_question": 6 // This is an independent question. Regardless of any other factors, only look for which image among all those provided captures the object {targetclass} most clearly. If none is found, return -1.
}
Now start the task: There are {num_view_selections} images for you to select from.

---

**memory_retrieval_prompt_for_verifiable_queries**

---

Imagine that you are in a room and tasked with finding a specific object. You already know the query content: {query}, the anchor object class: {anchorclass}, and the target object class: {targetclass}. The provided images are obtained by extracting frames from a video. Your task is to analyze these images to locate the target object described in the query.

You will receive multiple images, each with an ID marked in the upper left corner to indicate its order in the video. Adjacent images have adjacent IDs. Note that, to save space, multiple images may be combined and displayed together. You will also be given the query statement and a parsed version specifying the target object class and conditions.

Your task is divided into two main steps:

Step 1: Based on the query and associated conditions, determine whether any of the provided images contain the target object that satisfies the requirements. If found, return the corresponding image ID; if not, return -1.

Step 2: If no matching image is found in Step 1, ignore the query content and examine all images to see if any clearly capture an object of class {targetclass}. If such an image exists, return its ID; otherwise, return -1.

Please note that the query statement and conditions may not be fully satisfied in a single image, and they may also contain inaccuracies. Your goal is to find the object that most likely satisfies the query. If multiple candidates exist, choose the one you are most confident about.

Your response should be a JSON object containing the following fields:

{
"reasoning": "Your reasoning process", // Explain how you judged and located the target object. If cross-image reasoning is used, specify which images were involved and how.
"find_or_not": true, // Return true if a suitable image matching the query is found, otherwise return false.
"target_image_id": 4, // Return the image ID that best satisfies the query and conditions. If none found, return -1.
"anchor_image_id": 6, // Return the ID of the image where the anchor object is most clearly visible.
"extended_description": "The target object is a red box located in the lower left corner of the image.", // Describe the target object in the selected image, focusing on color and position.
"unique_question": 6 // This is an independent question. Regardless of other factors, select the image that most clearly captures an object of class {targetclass}. If none, return -1.
}

Now start the task:
There are {num_view_selections} images for your reference.
The following are the conditions for the target object: {condition}

**oss_prompt_for_unverifiable_queries**

Imagine that you are in a room and tasked with finding a specific object. You already know the query content: {query}, the anchor object class: {anchorclass}, and the target object class: {targetclass}. The provided images are frames extracted from a video in which the camera performs a full 360-degree rotation around a specific point. Your task is to analyze these images to locate the target object described in the query.

You will receive multiple images, each with an ID marked in the upper left corner indicating its sequence in the video. Adjacent images have adjacent IDs. To save space, multiple images may be combined and displayed together. Additionally, you will be provided with the query statement and its parsed version, which specify the target class and grounding conditions.

Your goal is to find the image that most clearly and completely captures the target object described by the query. The conditions may not be fully accurate or verifiable from a single image, so the correct object may not satisfy all of them. Try your best to identify the object that most likely meets the conditions. If multiple candidates appear correct, choose the one you are most confident about.

While checking each image, consider different views throughout the 360-degree rotation. If you find the target object in an image, also examine whether other images capture the same object more clearly or completely, and return the best one. Your answer should be based on the image where the target object is most clearly and completely visible.

Please reply in JSON format, structured as follows:

{
"reasoning": "Your reasoning process", // Explain the process of how you identified and located the target object. If reasoning across multiple images is used, explain which images were referenced and how.
"target_image_id": 1, // Replace with the actual image ID (only one) that most clearly captures the target object.
"reference_image_ids": [1, 2, ...], // A list of image IDs that also contain the target object and helped in reasoning.
"extended_description": "The target object is a red box. It has a black stripe in the middle.", // Describe the target object's appearance based on the selected image. Color and features only; do not include position.
"extended_description_withposition": "The target object is a red box located in the lower left corner of the image." // Describe the target object with both appearance and spatial position in the image.
}

Now start the task:
There are {num_view_selections} images for your reference.
Here is the condition for the target object: {condition}

**oss_prompt_for_verifiable_queries**

Imagine that you are in a room with the task of finding specific objects. You already know the query content: {query}, the anchor object category: {anchorclass}, and the target object category: {targetclass}. The provided images are extracted frames from a video that rotates around a certain point. Each image is marked with an ID in the top-left corner to indicate its sequence in the video. Adjacent images have adjacent IDs. For space efficiency, multiple images may be combined and displayed together.

You will also receive a parsed version of the query, which clearly defines the target object category, the anchor object category, and grounding conditions.

Your task consists of the following three steps:

Step 1: Based on the anchor object category, determine whether any of the provided images clearly capture the anchor object. If no such image is found, return -1 directly.

Step 2: If Step 1 is successful, return the smallest image ID (denoted as min_ID) among the images that clearly capture the anchor object.

Step 3: Among the images with IDs from 0 to min_ID, try to find an image that clearly captures the

target object and satisfies the query content and conditions. If such an image is found, return its ID; otherwise, return -1.

Note: The query statement and conditions may not be perfectly accurate or fully visible in a single image. Try your best to locate the object that is most likely to match these conditions. If multiple objects are plausible, select the one you are most confident about.

Here is an example: In Step 1, images 12, 13, 14, and 15 all clearly capture the anchor object, so Step 2 yields min_ID = 12. In Step 3, no image from ID 0 to 12 meets the query requirements, so target_image_id = -1.

Please reply in JSON format as follows:

{
"reasoning": "Your reasoning process", // Explain the reasoning process across all three steps. If cross-image reasoning is involved, specify which images were used and how.
"anchor_image_id": 12, // Return the smallest image ID that clearly captures the anchor object. If none is found, return -1.
"target_image_id": 4, // If anchor_image_id = -1, then return -1 directly. Otherwise, return the image ID ($\leq$ anchor_image_id) that best satisfies the query. If none found, return -1.
"extended_description": "The target object is a red box located in the lower-left corner of the image.", // Describe the target object in the image with ID = target_image_id. If target_image_id = -1, return None.
"unique_question": 6 // This is an independent question. Regardless of other factors, return the ID of the image that most clearly captures an object of class {targetclass}. If none found, return -1.
}

Now start the task:
There are {num_view_selections} images for your reference.
Here are the conditions for the target object: {condition}

## relation_parsing_prompt

You are an agent who is highly skilled at analyzing spatial relationships. You are given a query: {query}, a target object: {classtarget1}, and an anchor object: {anchorclass}. Your task is to determine the spatial relationship of the target object relative to the anchor object based on the query content.

The possible spatial relationships are defined as follows:
- up: the target object is above the anchor object // the target object is lying on the anchor object // the target object is on top of the anchor object.
- down: the target object is below the anchor object // the target object is supporting the anchor object // the anchor object is on top of the target object.
- near: the target object is close to the anchor object.
- far: the target object is far from the anchor object.
- between: the target object is between multiple anchor objects.

Please reply in JSON format with one key, "reasoning", indicating the spatial relationship you determine:

{
"reasoning": "up" // Return the spatial relationship type (up, down, near, far, or between) that best describes the position of the target object relative to the anchor object.
}

Now start the task.

## sras_choose_target_prompt

Imagine you're in a room tasked with finding a specific object. You already know the anchor object class: {anchorclass}, the target object class: {targetclass}, and the query the target object should match: {query}. The provided images are captured during a 360-degree rotation around the anchor object.

You are given a sequence of indoor-scanning video frames and a query describing a target object in

the scene. Your task is to analyze the images and locate the target object according to the query content.

Each image is annotated with an ID in the top-left corner indicating its sequential position in the video. Adjacent images have adjacent IDs. For space efficiency, multiple images may be combined and displayed together. You are also provided with a parsed version of the query, which lists the conditions that the target object should satisfy.

After filtering and comparison, your goal is to identify the image ID that contains the target object most clearly based on the query and conditions. Note that these conditions may not be fully observable in a single image and might be imprecise. The correct object may not meet all conditions. Try to find the object that most likely satisfies them. If multiple candidates seem plausible, choose the one you are most confident about. If no object meets the query criteria, make your best guess. Usually, the target object appears in several images—return the one where it is captured most clearly and completely.

Please reply in JSON format with the following structure:

{
"reasoning": "Your reasoning process", // Explain how you identified and located the target object. If you used multiple images, describe which ones and how they contributed to your decision.
"target_image_id": 1, // Replace with the actual image ID that most clearly shows the target object. Only one ID should be provided.
"reference_image_ids": [1, 2, ...], // A list of other image IDs that also helped confirm the target object's identity.
"extended_description": "The target object is a red-colored box. It has a black stripe across the middle.", // Describe the target object's color and notable features. No need to mention its position.
"extended_description_withposition": "The target object is a red-colored box located in the lower left corner of the image." // Describe both appearance and position of the object in the selected image.
}

Now start the task:
There are {num_view_selections} images for your reference.
Here is the condition for the target object: {condition}

**compare_prompt**

You are an intelligent assistant who is extremely proficient in examining images. You already know the target object category: {target_class}. Now I will provide you with two images. You need to determine whether the target objects captured in these two images are in the exact same position. Since these two images are taken from the same pose, you only need to check whether the target objects are in the same position within the images.

For example, if the target object is a table and you can clearly see that the table is located in the middle of both images, then the target objects captured in these two images are considered to be in the same position.

Please reply in JSON format with two keys: "reasoning" and "images_same_or_not":

{
"reasoning": "Your reasons", // Explain the basis for your judgment on whether the target objects captured in these two images are in the same position.
"images_same_or_not": true // It should be true if you think the target objects captured in the two images are in the same position. If you find that the positions of the target objects captured in the two images are different, or if the target object is captured in the first image but not in the second, then it should be false.
}

**fallback_prompt**

Imagine you are in a room tasked with finding a specific object. You already know the query content: {query}, and the target object category: {targetclass}. The images provided to you are frames extracted from a video that rotates around a particular point. Each image is marked with an ID in the top-left corner to indicate its sequence in the video, and adjacent images have consecutive IDs. For space efficiency, multiple images may be combined and displayed together.

Your task consists of two steps:

Step 1: Locate an image that contains the target object that satisfies the query statement and its associated conditions. The image must clearly and completely capture the target object. If such an image is found, return its ID and skip Step 2.

Step 2: If no image meets the query-based requirements, ignore the query and check all provided images. Identify an image that clearly captures the object of category {targetclass}. If such an image is found, return its ID. If none are found, return -1.

Please reply in JSON format with the following structure:

{
"reasoning": "Your reasoning process", // Explain the reasoning behind both steps of your decision-making process.
"match_query_id": 12, // Return the image ID that satisfies Step 1. If no image matches the query, return -1.
"object_image_id": 4, // If Step 1 is successful, return -1 here. Otherwise, return the ID of the image that clearly captures the object in Step 2. If not found, return -1.
"extended_description": "The target object is a red box located in the lower-left corner of the image." // Provide a brief description of the target object as seen in the selected image. Focus on visual features such as color and location within the image.
}

Now start the task:
There are {num_view_selections} images for your reference.

**get_good_view_prompt**

You are an excellent image analysis expert. I will now provide you with several images, each marked with an ID in the upper left corner. These images are captured by rotating around a target object {target} that is framed with a green bounding box in the reference image. The reference image is also provided, and it contains the target object {target} enclosed by a green box, with the word "refer" shown in red in the upper left corner.

Your task is to determine which three (at most four) of the provided images capture the target object from the reference image most clearly and completely. Please note that, for layout efficiency, multiple images may be displayed together in a single composite image.

Your response should be in JSON format, containing the following fields:

{
"reasoning_process": "Your reasoning process", // Explain how you select the images that best capture the target object framed in the reference image.
"image_ids": [2, 4, 5, 7] // Replace with the actual image IDs. Return up to four IDs corresponding to the images that, in your opinion, capture the target object most clearly and completely.
}

Now start the task:
There are {num_images} candidate images and one reference image for you to choose from.

**bboxchoose_prompt**

Great! Here is the detailed version of the picture you've selected. There are {num_candidate_bboxes} candidate objects shown in the picture. I have annotated an object ID at the center of each object with white text on a black background. You already know the query content: {query}, the anchor object: {anchorclass}, and the target object: {classtarget}. In addition, you will be provided with an extended description: {description}, which includes the position of the target object in the picture.

Your task consists of two main steps:

Step 1: The candidate objects shown in the picture are not necessarily all of the target class {classtarget}. You must first determine which of them belongs to the class {classtarget}.

Step 2: Among the identified candidate objects of class {classtarget}, select the one that best matches both the query content and the extended description (including position).

Please reply in JSON format with two fields:

{
"reasoning": "Your reasoning processing", // Describe your full reasoning process in three parts: (1) how you identified candidate objects of the target class; (2) how you verified them against the extended description; and (3) how you selected the final object ID.
"object_id": 0 // The object ID you select. Always provide one object ID from the picture that you are most confident about, even if you think the correct object might not be present.
}

Now start the task: There are {num_candidate_bboxes} candidate objects in the image.

**limit_prompt**

Great! Now you will perform an expert judgment on the visibility of a target object in the provided image.
You already know the target object category: {targetclass}. You will be shown one image containing this object class.
Your task consists of two main steps:
Step 1: Some object categories, such as beds, sofas, closets, cabinets, shelves, etc., are considered inherently large. If the target object belongs to this group of large categories, directly return `"limit": true` without proceeding to the next step.
Step 2: If the target class is not considered large, examine the image and determine whether the target object appears to be fully captured. If you believe the object is incomplete or partially outside the frame, return `"limit": true`; otherwise, return `"limit": false`.
Please reply in JSON format with two fields:

{
"reasoning": "Your reasoning process", // Describe your reasoning clearly: (1) whether the category is considered large, and (2) if not, how you judged the completeness of the object in the image.
"limit": false // Return true only if the object is large, or if it is not large but appears incomplete in the image.
}

Now start the task: You are given one image and the target object category: {targetclass}.

## M    COST CALCULATION FOR METHODS

Before we officially begin, let us once again emphasize that all costs are reported in units of 1,000 seconds (e.g., 9k = 9000s). The results shown in tables ( Table 2, Table 3, Table 4, Table 5, Table 6, Table 7, Table 8, Table 15) have all been processed with unit normalization.

For both the baseline methods and our proposed MCG approach, the robot's initial camera pose is assumed to be at the center of the room (see the main text for the formal definition of this key assumption). For MCG, the full camera trajectory starts from the initial pose and follows a sequence of new poses generated by the MCG pipeline. The cost of the entire trajectory is computed according to the evaluation metrics defined in the main paper. For the WG and CRG baselines, all images are pre-captured and sequentially indexed. We first identify the image whose pose is closest to the initial camera pose and denote its index as $n$. The camera trajectory then starts from the initial pose and proceeds through the poses of images with indices $n, n + 1, n + 2, \ldots$, wrapping around from the last index back to 1 as needed, and ending at index $n - 1$. The cost is computed based on the same evaluation procedure. For the MOG baseline, which only utilizes memory images, the camera trajectory consists of only two poses: the initial pose and the pose of the target image. Its cost is similarly computed using the defined metrics.

## N    OPEN PROBLEMS

We present the ChangingGrounding benchmark (CGB) as the first benchmark for evaluating 3D visual grounding in changing scenes and introduce the Mem-ChangingGrounder (MCG) as a strong baseline method. Nevertheless, both still exhibit the following limitations.

### N.1  Limitations of the CGB Benchmark

At present, our CGB dataset models only the relative positional changes between the target and its surroundings, without accounting for critical factors such as lighting variations, object appearance attributes (e.g., color, material, deformation), or dynamic scene interactions. Moreover, its repertoire of spatial relations lacks allocentric descriptions like "Object A is in front of Object B." These omissions narrow the benchmark's breadth and depth when assessing an agent's cross-scene generalization and robustness. Future work can address these gaps by enriching multimodal annotations, introducing additional dimensions of variation, and incorporating allocentric relations, thereby expanding the dataset's scale and diversity and enhancing CGB's applicability and challenge in real-world dynamic environments.

### N.2  Limitations of the MCG method

**Limitations of VLM capability.**  MCG relies heavily on the underlying Vision–Language Model (VLM) to locate target objects in image sequences according to the analysis requirements. As demonstrated by the ablation studies above, the strength of the VLM has a decisive impact on MCG's final grounding accuracy. If the VLM is insufficiently capable—or if the visual information in real-world scenes is unusually complex—MCG's performance can deteriorate. Nevertheless, because VLM technology is advancing rapidly, we can replace the current module with more powerful models in the future to further enhance performance.

**Noise from rendered images.**  During the experiments, MCG consistently feeds rendered RGB-D images into the vision-language model (VLM) for inference or uses them for SAM-based segmentation, projection, and related processes. However, the rendering process based on mesh files introduces various types of noise, including artifacts in the RGB images and inaccuracies in the depth maps. Moreover, there may be inherent differences in how VLMs process real versus rendered images. These factors can negatively affect the grounding accuracy.

**Noise introduced by 2D models.**  MCG depends on 2-D object detectors and segmentation networks to filter candidate images and perform the final projection. Although state-of-the-art models such as GroundingDINO and SAM are highly capable, they still exhibit missed detections, false positives, imprecise bounding boxes, and segmentation errors. These imperfections propagate through the pipeline and ultimately undermine the accuracy of the grounding results.

**Future work** Despite these limitations, we believe that our work on MCG and the CGB benchmark provides a strong foundation for future research in the field of grounding tasks in changing scenes. We hope that our contributions will inspire researchers to explore new methods and techniques to address the challenges posed by dynamic scenes. Specifically, we encourage the community to focus on the following open problems: (1) Improving VLM Robustness: Developing more robust Vision–Language Models that can handle complex real-world visual information and reduce the impact of noise; (2) Enhancing Multimodal Integration: Exploring ways to better integrate multimodal data (e.g., combining visual, linguistic, and spatial information) to improve grounding accuracy; (3) Expanding Benchmark Diversity: Contributing to the expansion of the CGB benchmark by adding more diverse scenarios, including variations in lighting, object appearance, and dynamic interactions; (4) Reducing Noise in Rendered Data: Investigating methods to minimize the noise introduced during the rendering process and to bridge the gap between real and rendered images; (5) Advancing 2D-to-3D Projection Techniques: Improving the accuracy and reliability of 2D object detection and segmentation models to enhance the overall grounding performance. We hope that our work will serve as a catalyst for further research in this exciting and challenging domain. By addressing these open problems, we can collectively push the boundaries of 3D visual grounding in changing environments and develop more effective and robust solutions.

## O  More results

### O.1  Test Samples

For each sample, first, we select any reference scan as $S_c$ and randomly select one rescan of $S_c$ as $S_p$. We then randomly pick an object $O$ with descriptions $D^o$ in $S_c$ as the target object and user query. It is important to note that, in order to ensure that the test samples cover diverse types of descriptions,

we selected a fixed number of instances from every relation type. Within the 250 samples, both the anchor object and the target object may either remain static or undergo changes.

## O.2 RENDERED VS. REAL IMAGES IN MEMORY

In previous experiments, both the memory and exploration images used by our system were re-rendered images. We still don't know how well VLMs work with these synthetic images. To check this, we conduct a comparative experiment with two settings. In the w.rendering setting, both the memory and exploration inputs are re-rendered images, consistent with the main experiment. In the w/o.rendering setting, the exploration images remain rendered, while the memory images are replaced with real photographs. Note that we don't have real images captured with the unified camera module described in the main text Section 3.3. To align our rendered images with the real photos supplied by 3RScan, we render every image using the original camera model from 3RScan.

We randomly sample 50 instances from a pool of 250 and observe the final grounding results. As shown in Table 15, experimental findings indicate that using rendered images in memory does not significantly affect the overall grounding accuracy. Results show that the w.rendering setting appears to perform slightly worse than the w/o.rendering setting. That does not prove that rendering is superior because there exists normal experimental variance. Moreover, the MCG pipeline still requires many exploration images that must be rendered for VLM inference. Overall, these results suggest that using rendered images in our experiments is a feasible approach.

Table 15: Comparison between using rendered and real images in memory.

| Version | Acc@0.25 | A_c | M_c |
|---|---|---|---|
| w. rendering | 28 | 1.74 | 2.05 |
| w/o. rendering | 24 | 1.62 | 1.94 |

Table 16: Human result.

| Method | Acc. | A_c | M_c |
|---|---|---|---|
| Human | 85.6 | 44.23 | 17.51 |

## O.3 HUMAN ACCURACY

We additionally asked a human researcher to perform grounding on the same set of 250 test examples, following the same general procedure as in the WG setting. Unlike the MCG and baseline models, whose accuracy is determined by the IoU-based matching criterion, human performance is evaluated solely based on whether the correct target can be successfully identified. As shown in Table 16, it is worth noting that human performance remains substantially higher than the MCG and all baselines, indicating that the ChangingGrounding task still holds substantial room for further research and improvement.

## O.4 DETAILED DISCUSSION REGARDING VLMS

To investigate the primary ways this system shows brittleness in different ways of prompting the underlying VLMs. We study two types of memory retrieval and grounding prompting modification, language modification, and vision modification. For language modification, we remove the parts requiring reasoning in the original prompt and simplify the detailed search instructions. For vision modification, we change the image-stitching strategy in the original prompt from dynamic stitching to a fixed 2×4 layout. In Table 11, v_origin represents the original prompt, v_less represents the language modification, and v_fix_layout represents the vision modification. The results show that

v_less leads to only a slight decline in accuracy, whereas v_fix_layout causes a more noticeable performance drop because, as the number of images increases, the fixed layout produces more stitched images than dynamic stitching, making the grounding task more challenging for the VLM.

To investigate how different underlying LLM/VLMs affect the performance of MemChangingGround. We studied 3 types of VLMs, gpt ( GPT-4.1 (OpenAI, 2025a) ), gemini ( Gemini-2.5-Flash (Comanici et al., 2025)) and claude ( Claude-Sonnet-4.5 (Anthropic, 2025) ). As shown in Table 12, gpt exhibits the best overall performance and provides the most stable formatting consistency, followed by gemini with slightly lower accuracy. claude shows a more significant performance reduction, which we found is largely due to its difficulty in adhering to the required output format, consequently affecting its final accuracy.

Finally, to investigate how VLMs uncertainty would influence the MCG results, we increased the VLM's temperature and top-p settings. As shown in Table 13, even under higher-temperature sampling settings, the accuracy and constraint metrics remain close to the original configuration, indicating that increasing output diversity and randomness does not substantially affect the overall performance.

## O.5   3D Visual Grounding Methods Implementation

Prior 3D visual grounding methods are not readily adaptable to scenarios involving dynamic visual grounding because they are not designed to leverage memory. These methods typically require the latest point clouds of the current scene for each grounding instance, which is impractical for real-world applications since rescanning the entire scene to obtain updated point clouds is highly inefficient. Nevertheless, we attempt to adapt these methods to incorporate memory for our task setting.

We designed a pipeline as follows: the model initially uses point clouds from memory (past scenes) to locate the anchor object based on the query. Once the anchor object's position is identified, the model determines a neighboring region around the anchor object to obtain updated point clouds. This approach eliminates the need for scanning the entire scene. The neighboring region is defined as within a 2-meter radius of the anchor object's position.

For cost calculations, we make an approximation based on the assumption that the agent starts from the center of the room, moves to the previously predicted anchor object location, and performs a full 360-degree rotation around the vertical axis to scan the region. It is important to note that this assumption is also not always feasible in real-world scenarios. Specifically, a single 360-degree rotation at one position often cannot capture all details, resulting in estimated costs that are significantly lower than the actual costs.

We conducted experiments using the 3D-Vista (Zhu et al., 2023) model, as it is pre-trained on the 3RScan dataset. It should be noted that this model requires pre-detection of all 3D bounding boxes prior to grounding. For our experiments, we utilized GT bounding boxes, which significantly enhance performance beyond realistic scenarios.

The final experimental results are presented in the Table 8. We create the pipeline as follows: the agent first uses 3D-Vista to locate the anchor object based on the query in the point cloud of a previous scene. After obtaining the position of the anchor object, the agent uses this position as the center to crop a region within a 2-meter radius from the current scene's point cloud. This region is then provided as input to 3D-Vista for inference, under the same assumption that 3D-Vista has access to the ground-truth bounding box contained within this region.

Please note that these experiments are intended solely for reference. We do not consider them to have practical significance due to simplifying assumptions. Specifically, at Acc@0.25, our method demonstrates superior accuracy and lower action costs. Additionally, since 3D-Vista performs a full 360-degree rotation during scanning (an impractical scenario), it exhibits nearly zero translation cost and reduced rotation cost.

## O.6   Inference time

**Success rate.** Specifically, A step for modules in the MRGS is counted as successful under 2 following conditions: (a) View Pre-selection (VP): The pre-selected views from the SRAS or the

OSS contain the target object. (b) VLM Choose Image (VCI): The VLM predicts an image that contains the target object. A step for modules in the MS is counted as successful under 3 conditions: (a) OV Detection (OD): At least one detection box contains the target object. (b) Select Reference Instance (SRI): The VLM selects the detection box containing the target object. (c) Multi-view Projection (MP): The 3D IoU between the predicted box and the ground-truth box is $\geq 0.25$. Their success rate is in Table 9

**Inference time.** The table below shows the time consumption of modules in our framework.

Table 17: Inference time of different modules (unit: seconds)

| Query Analysis | View Preselection | Detection Predictor |
|---|---|---|
| 0.8 | 11.3 | 0.2 |
| SAM Predictor | Dynamic Stitching | Project Points |
| 0.5 | 9.7 | 0.7 |
| VLM Select Box | VLM Analysis (Spatial relations) | VLM Analysis (Static) |
| 11.4 | 1.0 | 5.4 |
| VLM Predict Images | VLM Choose Good Views | Depth Denoising |
| 11.7 | 11.5 | 0.7 |
| VLM Decide Distance Limitation | | **Total** |
| 4.9 | | **113.2** |

We acknowledge that the inference speed has significant room for optimization, given that this is a research project. For example, as shown in the Table 17, the majority of the time in our pipeline is spent on VLM inference. However, with the rapid advancement of VLM technology, we expect that high-speed large models will soon be deployable on edge devices such as the FastVLM project introduced by Apple (Vasu et al., 2025). FastVLM reaches 85× faster TTFT when compared specifically to LLaVa-OneVision operating at 1152×1152 resolution. This opens up promising opportunities for significantly reducing the overall runtime of our method.

## O.7    ERROR CASE ILLUSTRATION

In this section, we present concrete failure cases in the MCG framework.

**Wrong images for VLMs final predicting** First, owing to the VLM's capacity, it may fail to identify the anchor object in memory images at the beginning, causing errors in the whole grounding process. Once the anchor is wrong, new views from SRAS or OSS often miss the target. Second, views from SRAS and OSS may produce limited viewpoints that lack the correct object (anchor or target), especially when the correct object is located low. Third, there exist a lot of unusable rendering images, which often have large blank or missing regions. In all cases, they lead to a single situation where the VLM can't get any images containing the target object at all, which will cause failure for the final VLM grounding steps. There are some examples shown in Figure 5 and Figure 6.

**VLMs failure in grounding target images** Relational queries involving horizontal spatial reasoning (e.g., "select the chair closest to the table") impose higher demands on the inference capability of vision-language models (VLMs). Such relationships require the model to make fine-grained comparisons based on relative spatial distances rather than absolute object properties. In cluttered scenes, distractors with small distance gaps increase errors, making VLMs prone to wrong selections. An example is shown in Figure 7.

**Failure in SAM and projection** During our experiments, we observed that SAM often produces noisy masks by including pixels unrelated to the target, which we believe may stem from SAM's poor generalization to rendered images and the low quality of these images. This over-segmentation reduces the accuracy of 3D projection and bounding box localization. In addition, since our experiments are conducted on rendered images, missing or incomplete regions often affect precision. Although we applied denoising on depth maps by removing abrupt pixels, residual noise remains a challenge to accurate 3D localization. Examples are shown in Figure 8.

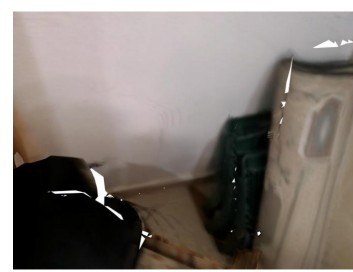 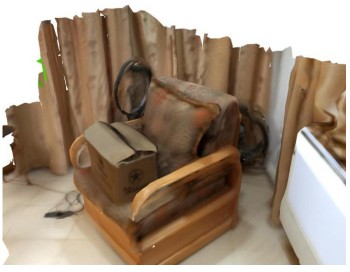

Wrong VLM choise           Ground truth

Figure 5: **VLMs failure in memory retrieval, the anchor object is a box.**

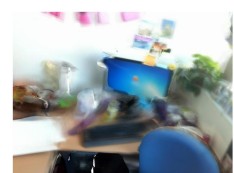 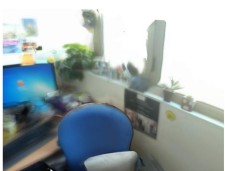 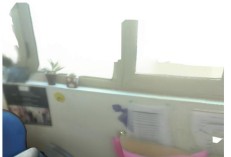 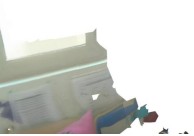

Wrong VLM choise           Correct

Figure 6: **Failure in SRAS, the user query is to find the cushion that is farthest from the pc.**

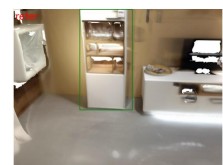 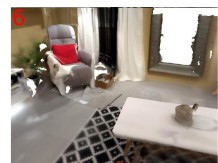 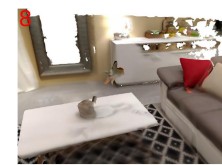 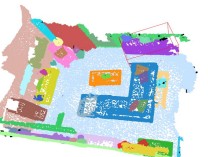

Wrong VLM choice      Anchor      Correct

Figure 7: **VLMs fail to ground the target image: query "cabinet near the box."**

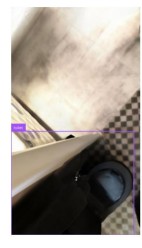 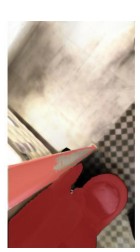 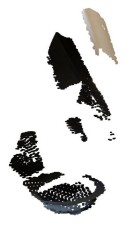 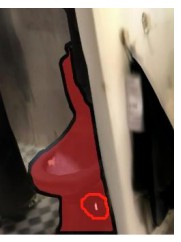 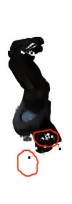

Wrong SAM result           Rendering noise

Figure 8: **Failure in SAM and projection.**

**Memory images preselection & Dynamic stitching**

**memory_retrieval_prompt_for_verifiable_queries**

"query": "choose the toilet that is under the flush"
"target_class": "toilet"
"anchor_class": "flush"

**Memory retrieve**

{'reasoning': "The query asks for a toilet that is under the flush, with the anchor object being the flush. I first scanned the images for the presence of a toilet and a flush. The clearest images showing both the toilet and the flush are those in the range of IDs 006 to 076. In these images, the flush (the rectangular panel above the toilet) is visible directly above the toilet, which matches the condition 'under the flush.' Among these, image 006 provides a clear view of both the flush and the toilet, with the flush directly above the toilet, making it easy to confirm the spatial relationship. For the anchor object (flush), image 006 also provides a clear view, but image 081 shows the flush even more clearly, as it is more centered and less obstructed. For the unique question, the toilet is most clearly visible in image 073, where the entire bowl is unobstructed and well-lit.", 'find_or_not': True, 'target_image_id': 6, 'anchor_image_id': 81, 'extended_description': 'The target object is a white toilet located in the lower right corner of the image, with the flush panel directly above it on the wall.', 'unique_question': 73}

**compare_prompt**

"target_class": "flush"

**Stay static or not**

{
"reasoning": "In both images, the flush (the rectangular button panel above the toilet) is located in the same position on the wall, just above the toilet seat and below the two round meters and a small round sign. The orientation and placement relative to the other objects (such as the mirror, sink, and wall outlet) are consistent between the two images. There is no noticeable shift or change in the position of the flush within the frame.",
"images_same_or_not": true
}

**compare_prompt**

"target_class": "toilet"

**Stay static or not**

{
"reasoning": "In both images, the toilet is located in the lower right corner of the image, adjacent to the bathtub and near the same set of objects (such as the toilet paper holder and cleaning supplies). The angle and position of the toilet relative to the other bathroom fixtures are consistent between the two images. The only noticeable difference is the toilet seat cover design, but the position of the toilet itself remains unchanged.",
"images_same_or_not": true
}

**limit_prompt**

"target_class": "toilet"

**Big stuff or occluded stuff**

{'reasoning': 'Step 1: The target object is a toilet, which is not considered a relatively large object like a bed, sofa, or cabinet, so proceed to Step 2. Step 2: In the provided image, only a portion of the toilet is visible, with the seat and part of the bowl shown, but the full structure (such as the tank, base, and full outline) is not completely captured. Therefore, the toilet is not photographed completely.', 'limit': True}

Figure 9: **Case of the MCG grounding part-1.**

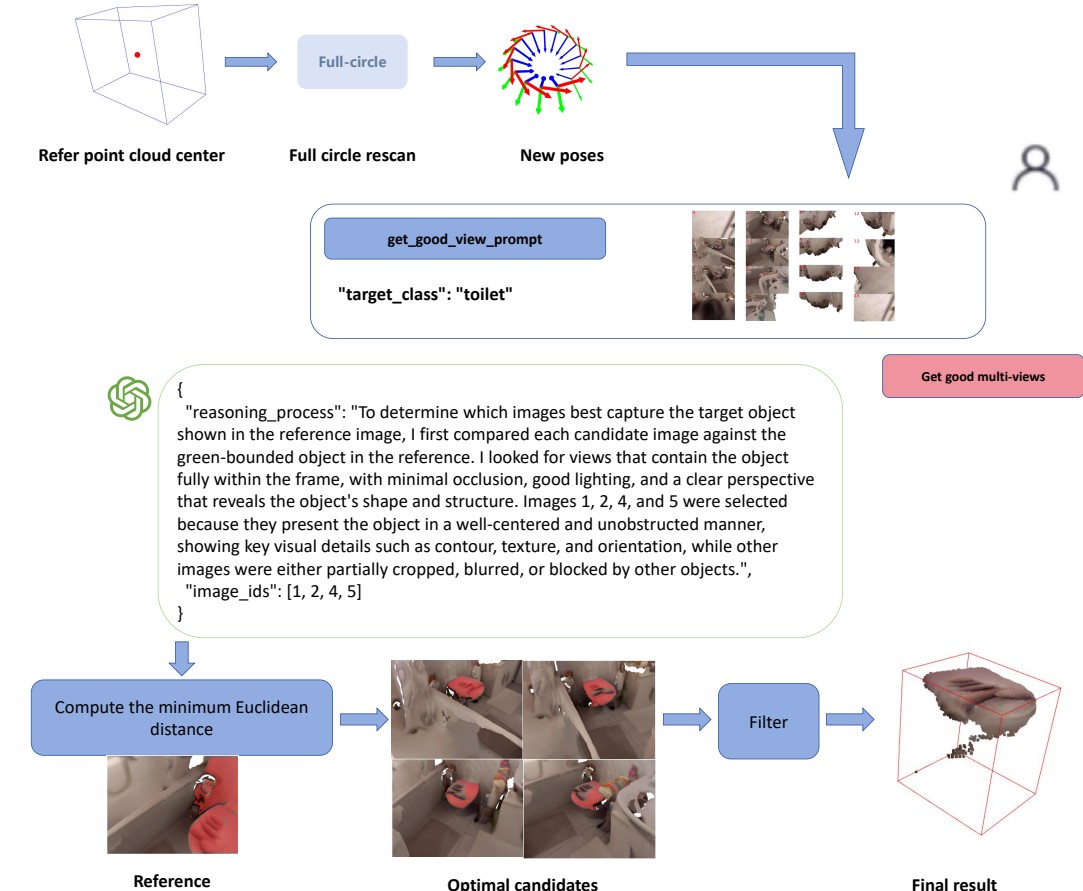

Figure 10: **Case of the MCG grounding part-2.**

## O.8 FULL DEMO

In this section, as shown in Figure 9 and Figure 10, we present a representative and structurally concise example to intuitively illustrate how MCG effectively leverages memory information for efficient and accurate target localization in dynamic environments. The example highlights the central role played by the vision-language model (VLM) throughout the entire execution process. We provide a detailed depiction of the VLM's reasoning at each step, demonstrating how it progressively converges on the target object through multi-round perception and decision-making, thereby showcasing its capabilities in semantic understanding and spatial reasoning.

