# OpenReview forum: "ChangingGrounding: 3D Visual Grounding in Changing Scenes"
_ICLR.cc/2026/Conference — Submitted to ICLR 2026_

### Official Review · Reviewer_WRSq · 2025-10-26

**Soundness:** 2
**Presentation:** 2
**Contribution:** 2
**Rating:** 2
**Confidence:** 4

**Summary:**

The authors propose ChangingGrounding, a new 3D visual grounding task that uses memory from past observations.
To evaluate the performance, the authors provide a dataset with corresponding evaluations.
The proposed zero-shot agent-based pipeline Mem-ChangingGrounder outperforms the baselines on this newly-proposed task.

**Strengths:**

1. The motivation to explore 3D visual grounding in changing scenes is novel and interesting.
2. The scope is comprehensive. The authors (i) propose the task, (ii) provide the dataset and evaluation for the new task, and (iii) compare against several baselines for this new task.

**Weaknesses:**

1. The motivation and task formulation in the introduction section require clearer presentation. In particular, the comparison shown in Figure 1 is not sufficiently explained. If the reviewer understand it correctly, it seems that current problem formulation is the previous setting (one full scan) + several new explored image demos, so that it could avoid repetitive rescan of entire scenes mentioned in the motivation (line 041). The authors should consider make better clarification compare with the previous setting.
2. The task formulation in Sec. 3.1 is not well-constructed. It's not clear what are in $S_p$ and $M_p$. Are $S_p$ only a set of images, or it also contains the old 3D object bounding boxes? Also, if the output $B$ is the predicted 3D bounding boxes, then the evaluation could only conducted on accuracy. Its not clear how to use bounding boxes to evaluate motion and action costs.
3. As the task if formulated as $⟨S_p, S_c, M_p, D_c⟩ → B$ (line 158), it is not clear why $S_p$ would help with the localization accuracy. It seems that the past memory could either be (i) the same as current views or (ii) different but has a wrong position. The correct location should be included in $S_c$. It's not intuitive why the accuracy of the Wandering Grounding baseline is worse (Table 2).
4. Other than the actions and memory costs, the proposed zero-shot pipeline heavily relies on VLM to do reasoning and inference. From table 10, it seems the inference cost is huge for the proposed method. How would the inference time compared with the traditional 3D method? The authors only show the action and motion cost comparison in Table 8. The inference time comparison is missing.
5. In the pipeline figure (Figure 3), it is not clear which images are from $S_p$ and which are from $S_c$. It is better to have the corresponding variable notations on the figure to help readers understand.

**Questions:**

Please refer to the weakness section.

During rebuttal, the reviewer would like to see the authors' responses of the weaknesses, including the following:

1. The further clarification on the motivation and problem formulation (Weakness 1, 2, 3).
2. Additional experimental results analysis (Weakness 3).
3. Discussion and results on inference time comparison (Weakness 4).
4. A clearer explanation of the pipeline (Weakness 5).

The reviewer will update the evaluation after seeing the clarification from the authors.

---

> ### Author Response · Authors · 2025-11-26
> **Author Response to Reviewer WRSq (1/3)**
>
> Dear Reviewer WRSq,
>
> Thank you for the thoughtful feedback. We appreciate the recognition that our **motivation to study 3D visual grounding in changing scenes is novel and interesting**, and the acknowledgment of our **comprehensive scope**—(i) proposing the task, (ii) releasing a dataset with an evaluation protocol, and (iii) comparing against multiple baselines. We respond to each comment in detail below.
>
> ---
>
> **Weakness 1: Lack of a clearer presentation of motivation and task formulation.**
>
> **A1:**
> Thank you for your feedback.
>
> It is important to clarify that our setting is not simply replacing the previous approach with a full scan plus a few demonstrations. As described in Section 1, our framework relies on a **memory + exploration** mechanism to actively ground the target. In our setting, the agent needs to leverage its memory and exploration strategy to take actions and execute actions to obtain new observations about the scene. This differs fundamentally from the previous setting, where the task was executed within a fully **pre-scanned, fixed** scene.
>
> More specifically, as detailed in Sections 1 and 3, our **ChangingGrounding** setting differs from prior work in terms of its goals, requirements, and evaluation metrics:
>
> * **Goal**: The target that needs to be located is in an **environment that has already changed**, rather than in a static scene.
> * **Requirements**: The method **determines and executes actions to acquire new observations**, rather than relying on a fixed set of pre-scanned observations. In other words, the previously scanned scene, as memory, serves as priors for efficient exploration in the current scene. In our novel benchmark, we emphasize memory-based active exploration in a changed scene, while previous settings use an existing, static, fixed, pre-scanned scene in a passive perception manner.
> * **Evaluation**: The evaluation metric considers **both accuracy and cost**, instead of accuracy alone.
>
>
>
>
> **Weakness 2: Confusion about the task variation and cost calculation**
>
> **A2:**
> Sorry for the confusion.
>
> 1. **About $S_p$**:
>     In our formulation, $M_p$ denotes the **raw representation** of the previous scene, the RGB-D + pose sequence. For flexibility, we also allow methods for the ChangingGrounding task to leverage post-processed information derived from these raw representations (for example, a reconstructed point cloud of the previous scene based on the RGB-D + pose sequence). Therefore, we introduce the notation $S_p$ to **uniformly represent all information that can be derived or extended from the RGB-D + pose data**.
>
>     In our proposed MCG method, we do not explicitly use any additional derived information beyond the RGB-D + pose sequence, and thus, in this particular instantiation, we have $S_p = M_p$. But we still keep the more general notation $S_p$ to accommodate potential methods that may use this information in the future.
>
> 2. **About cost calculation**:
>     We provide a detailed explanation of how the cost is computed in Section 3.2 of the main paper. As we elaborated in our response to Weakness 1, ***our framework relies on a **memory + exploration** mechanism to actively ground the target. In our setting, the agent needs to leverage its memory and exploration strategy to take actions and execute actions to obtain new observations about the scene***. During this process, the sequence of actions executed by the agent, as well as the sequence of poses it reaches within the scene， are used to compute $C_a$ and $C_m$, following the precise formulations described in Section 3.2.
>
>
> **Weakness3: Confusion about the task definition and the results of the Wandering Grounding baseline.**
>
> **A3:**
> Sorry for the confusion.
>
> As we explained in our response to Weakness 1, in our approach, the agent obtains observations in $S_c$ based on actions guided by memory and exploration strategy. If memory is completely ignored and one blindly scans all of $S_c$ like the WG method, the observations become very redundant, and most of them are irrelevant to the target. This makes target localization more difficult.

---

> ### Author Response · Authors · 2025-11-26
> **Author Response to Reviewer WRSq (2/3)**
>
> **Weakness 4: Lack of direct time comparison and analysis.**
>
> **A4:**
> Thank you for your valuable feedback.
>
> We compare with 3D-Vista, a widely used traditional 3D visual grounding method. We add a **direct comparison** of task time with the 3D-Vista[1] here and will include it in the revised paper. It is worth noting that, in addition to comparing inference time, we also include a comparison of the total processing time. This is because traditional 3D methods take point clouds as input, and under our setting, the time from acquiring images to reconstructing the point cloud must also be considered. Below, we explain how we estimate the per-sample total processing time for 3D-Vista and MCG.
>
> For 3D-Vista, the total time consists of three parts: the time to acquire RGB-D images, the 3D reconstruction time, and the inference time. First, it needs to collect RGB-D images in the scene for 3D reconstruction; this part of the time is given by the average exploration cost $C_m$. Then, it performs 3D reconstruction based on these images, for which we approximate the cost using the time an advanced 3D reconstruction method ( e.g., VGGT[2] ) takes to process 100 images. Finally, we add the inference time of 3D-Vista itself. For MCG, we need to account for the average exploration cost $C_m$ and its inference time.
>
> The corresponding experimental results are shown below.
>
>
>
> | Method | Acc. | $C_a$ | $C_m$ | Inf(s) | T(s) |
> |--------|----------|----------|----------|--------|-----|
> | 3D-Vista | 33.2   | 18.00   | 2.39   | 0.05  |12.73  |
> | MCG (ours) | 36.8   | 8.47   | 9.76   | 113.2   |152.24   |
>
>
>
> The time reported for 3D-Vista[1] in the figure (12.73 s) consists of the average time to acquire RGB-D images (2.39k s / 250 = 9.56 s), the 3D reconstruction time (3.12 s), and the inference time (0.05 s). The time for MCG consists of the average exploration time (9.76k s / 250 = 39.04 s) and the inference time (113.2 s).
>
>
> As same as our analysis in Appendix O.3, we acknowledge that the MCG inference speed still has significant room for improvement, as this is a research project. As shown in Table 12, most of the time in our pipeline is spent on API Calling and VLM inference. With the rapid progress of VLM technology, we expect that high-speed large models will soon be available on edge devices. For example, the FastVLM project introduced by Apple[3] achieves an 85× faster TTFT compared with LLaVA-OneVision at 1152×1152 resolution. This progress opens promising opportunities for greatly reducing the overall runtime of our method.
>
> For 3D visual grounding methods such as 3D-Vista, they may rescan the entire scene each time to finish our changing grounding task, and the total time is shorter than the MCG method. However, the approach is still impractical. In dynamic environments, scenes are constantly changing, and it is unrealistic to perform a full rescan **every time** an object is moved. Worse yet, the system often does not know exactly whether, when the scene has changed, making it difficult to decide whether a new scan is necessary before grounding.

---

> ### Author Response · Authors · 2025-11-26
> **Author Response to Reviewer jGWF (3/3)**
>
> **Weakness 5: Lack of clear notations in Figure 3.**
>
> **A5:**
> Thanks for your suggestions.
>
> As described in Section 4 of our paper, the MCG pipeline contains two action policies within four core modules: Query Classification, Memory Retrieval and Grounding, Fallback, and Multi-view Projection. MCG first classifies the query, and then selects a path for memory retrieval and grounding based on the classification result. In the memory retrieval and grounding stage, MCG uses the information obtained from memory and explores the current scene with its action policies to locate the target. If this fails, the fallback module estimates the target. Finally, the multi-view projection module fuses the target obtained from the previous modules with the actively acquired multi-view observations around the target in the current scene for accurate grounding.
>
> In Figure 3, the leftmost box in the upper part illustrates the query classification and the preparation for memory image grounding. Therefore, all images in this box are from $S_p$. The second and third boxes from the left in the upper part show the memory retrieval and grounding processes under different types of queries. The anchors or target images grounded from the memory images are from $S_p$ (as explicitly shown by the table and microwave examples). However, the subsequent exploration images obtained in the current scene, as well as the final confirmed target image, all belong to $S_c$, although they are not explicitly displayed in the figure. The fourth and fifth boxes in the upper part correspond to the fallback process and the simplified multi-view projection process.
>
> The lower part of the figure illustrates the detailed procedures of the two action policies and the multi-view projection. Since both exploration and multi-view acquisition occur in the current scene, all images in these three boxes come from $S_c$.
>
>
> We have updated clear source annotations for images in Figure 3 in the revised version of the paper.
>
> ---
>
> **Q1: Further clarification on the motivation and problem formulation.**
>
> **A1:**
> Please refer to our response regarding Weakness 1, 2, 3.
>
> **Q2: Additional experimental results analysis.**
>
> **A2:**
> Please refer to our response regarding Weakness 3.
>
> **Q3: Discussion and results on inference time comparison.**
>
> **A3:**
> Please refer to our response regarding Weakness 4.
>
> **Q4: A clearer explanation of the pipeline.**
>
> **A4:**
> Please refer to our response regarding Weakness 5.
>
> [1]Ziyu Zhu, Xiaojian Ma, Yixin Chen, Zhidong Deng, Siyuan Huang, and Qing Li. 3d-vista: Pre-trained transformer for 3d vision and text alignment. In Proceedings of the IEEE/CVF International Conference on Computer Vision, pp. 2911–2921, 2023.
>
> [2]Jianyuan Wang, Minghao Chen, Nikita Karaev, Andrea Vedaldi, Christian Rupprecht, and David Novotny. Vggt: Visual geometry grounded transformer. In Proceedings of the Computer Vision and Pattern Recognition Conference, pp. 5294–5306, 2025b.
>
> [3]Pavan Kumar Anasosalu Vasu, Fartash Faghri, Chun-Liang Li, Cem Koc, Nate True, Albert Antony, Gokula Santhanam, James Gabriel, Peter Grasch, Oncel Tuzel, et al. Fastvlm: Efficient vision encoding for vision language models. In Proceedings of the Computer Vision and Pattern Recognition Conference, pp. 19769–19780, 2025.
>
> Thank you once again for your time!

---

> > ### Comment · Reviewer_WRSq · 2025-11-27
> >
> > Thanks for the detailed responses. Here are some follow-up comments and questions:
> >
> > For weakness1: I think the task formulation and the comparison against existing work is still not clear. In the rebuttal the authors were trying to explain with the specific methods/mechanisms, which is independent of the task formulation. Please provide a more straightforward comparison: what is the input/output for previous setting and what is the input/output for your proposed setting?
> >
> > For weakness2: The authors did not answer my question. In the current $\left\langle S_p, S_c, M_p, D_c\right\rangle \rightarrow B$ formulation, the output is only 3D bounding boxes, then the evaluation could only conducted on accuracy. To evaluate on costs, the action $a_t$ needs to be defined as part of the output in the problem formulation.
> >
> > For weakness3: The authors only addressed part of my questions. It is not clear why $S_p$ would help with the localization accuracy. It seems that the past memory could either be (i) the same as current views or (ii) different but has a wrong position.
> >
> > For weakness 4: Thanks for reporting the inference time comparison. Based on the results, it seems that re-do a scan and apply a 3D visual grounding method takes significantly less time while achieving the comparable results. The authors were arguing "the approach is still impractical" and "it is unrealistic to perform a full rescan every time an object is moved", but it seems applying the proposed method is much more expensive in terms of inference speed. Could the authors provide some evidence that the proposed setting/methods is more practical? For example, if one object is changed in the scene, how long would the proposed method takes to inference again based on the memory, and how long does traditional 3D grounding method takes?
> >
> > For weakness 5: Thanks for the explanation and the updated figure. It is easier for readers to understand now.
> >
> > I have also read the reviews from other reviewers. As this introduces a new benchmark/setting for 3D grounding, my major concern is that the current manuscript is not defining the problem setting clearly (weakness 1, 2, 3). Also, the proposed method seems to take much longer for inference, which somehow violates the motivation of the proposed setting (weakness 4). Method-wise designs should be further discussed after the problem setting is clear enough to the readers.

---

> > > ### Author Response · Authors · 2025-12-03
> > >
> > > Dear Reviewer WRSq,
> > >
> > > Thank you for your reply. We try to make things clearer as belows:
> > >
> > > **Q1: "Provide a more straightforward comparison: what is the input/output for previous setting and what is the input/output for your proposed setting?"**
> > >
> > > **A1:**
> > >
> > > **Previous setting:** point clouds of a static scene + a language query → 3D bounding box of the target object in the static scene.
> > >
> > > **Our proposed setting:** RGB-D sequences (and their derivatives) of a previous scene (optional) + a language query + RGB-D sequences (and their derivatives) of the current scene (optional) → 3D bounding box of the target object in the current scene.
> > >
> > > **Q2: "To evaluate on costs, the action needs to be defined as part of the output in the problem formulation."**
> > >
> > > **A2:** In our end-to-end formulation, the final objective is to predict the 3D bounding box of the target object, and **the action sequence is treated as an intermediate result rather than part of the task output. It is possible that a method relies solely on its memory and does not perform any action in the current scene**, yet can still predict the 3D bounding box. In this case, actions are not part of the output at all. **We therefore keep the formulation flexible and general by not including actions in the output definition.** However, whenever a method needs to actively explore the scene, we can still evaluate its action cost.
> > >
> > > **Q3: "Why $S_p$ would help with the localization accuracy."**
> > >
> > > **A3:** Without $S_p$, the baseline needs to wander around the entire scene to find the target without any prior knowledge. In this case, it may fail to observe the target object, or it may collect many redundant observations that are unrelated to the target, which hurts localization accuracy. In contrast, $S_p$ provides useful priors. **For example, a person can usually find an object much more easily in their own home than in an environment they have never visited before.**
> > >
> > > **Q4: "Could the authors provide some evidence that the proposed setting/methods is more practical?"**
> > >
> > > **A4:** Although our method appears to take more time for inference, most of the overhead comes from API calls to OpenAI’s ChatGPT. As a research prototype, we did not specifically optimize this part. However, **we expect that our approach will not be fundamentally limited by inference time: as computation continues to improve, on-device VLMs will become more powerful, and this overhead will be greatly reduced. We discuss this in Section 5.4 of the main paper.**
> > >
> > > Compared with traditional methods that need to re-scan the scene each time, the cost of scanning dominates the runtime and is typically much larger than that of our MCG. **If the system is deployed in a large multi-floor shopping mall, the scanning cost would grow even further. We do not think it is practical to re-scan the whole supermarket every time a customer asks the system to find an item.**

---

### Official Review · Reviewer_jGWF · 2025-10-29

**Soundness:** 3
**Presentation:** 3
**Contribution:** 2
**Rating:** 6
**Confidence:** 3

**Summary:**

This paper introduces a new setting for 3D Visual Grounding (3DVG), termed Changing Grounding, and proposes a corresponding baseline method called Mem-ChangingGrounder (MCG). The motivation is that traditional 3DVG assumes full-scene reconstruction in advance, which is unrealistic in embodied environments. In contrast, embodied agents must actively explore, incrementally reconstruct local regions, and rely on memory for reasoning. The proposed MCG baseline adopts an agent-based design, integrating memory retrieval, VLM-based analysis, and GroundingDINO + SAM + projection to obtain 3D bounding boxes. Experiments demonstrate its overall effectiveness.

**Strengths:**

1. The motivation is highly valuable, the proposed active, memory-enabled, and incremental reconstruction setting aligns well with emerging embodied scenarios, especially in robotic navigation.
2. The paper contributes a Changing Grounding dataset and introduces new metrics that balance grounding accuracy and exploration cost, which are both practical and meaningful.
3. The proposed baseline is comprehensive and functional, showing promising performance advantages in experiments.

**Weaknesses:**

1. The baseline (MCG) is an agent-based pipeline with considerable complexity, and obtaining 3D bounding boxes via 2D SAM + projection may introduce significant localization errors compared to direct 3D detection.
2. Only an agent-based baseline is provided; there are no learnable or fine-tuned baselines (e.g., Transformer-based models such as DETR-style 3D grounding).
3. The memory mechanism may face scalability issues in long-horizon tasks, as the current design lacks memory compression or forgetting strategies, potentially leading to high computational overhead.
4. There is a minor ambiguity in the accuracy metric, when multiple candidate bounding boxes exist, it is unclear how accuracy is computed or which box is selected.

**Questions:**

See Weaknesses.

---

> ### Author Response · Authors · 2025-11-26
> **Author Response to Reviewer jGWF (1/2)**
>
> Dear Reviewer jGWF,
>
> Thank you for the encouraging feedback. We appreciate the recognition that our **active, memory-enabled, incremental reconstruction** setting aligns with emerging **embodied scenarios**, the value of our **Changing Grounding** dataset and **metrics balancing accuracy and exploration cost**, and the strength of our **comprehensive baseline** with **promising performance**. We respond to each comment in detail below.
>
> ---
> **Q1: Concern about 2D SAM+ noise.**
>
> **A1:**
> Thank you for your thoughtful question.
>
> We acknowledge that 2D SAM[1] and related models indeed have certain limitations. However, we **have used effective mechanisms** in MCG to mitigate the noise and instability. On the other hand, 3D detections themselves also **suffer from inherent limitations**. We will explain these two aspects separately below.
>
> ### **Our strategies to address the limitations of 2D SAM+**
>
> 1. We incorporate a **multi-view projection module**, which aggregates projection results from multiple viewpoints. This reduces the incompleteness and bias that commonly occur when relying on a single-view projection.
>
> 2. Second, our method adopts noise-mitigation techniques for 2D segmentation[2], including:
>
>     * **Erosion** is applied to clean up small artifacts and slightly contract the mask edge, which helps avoid depth inaccuracies near object boundaries.
>
>     * **Connected-component filtering** is used to keep only the two most substantial regions in the mask, discarding spurious fragments while retaining the main valid parts of the object.
>
>
> ### **Limitations of 3D detection**
> 1. 3D detection typically relies on a **good reconstructed point-cloud scene before a 3D detector can be applied.** However, this requirement fundamentally **contradicts the goal** of the ChangingGrounding task, which aims for accurate and **efficient** localization in dynamic environments. Obtaining a high-quality reconstructed scene would **inevitably require extensive and redundant exploration**.
>
> 2. 3D detectors exhibit limited generalization. Most existing 3D detectors are trained on **annotated 3D datasets**, which are inherently **scarce**. As a result, when confronted with noisy reconstructed point clouds or entirely novel 3D environments, these detectors may fail to maintain reliable performance.
>
>
>
>
> **Q2: Lack of learnable or fine-tuned baselines.**
>
> **A2:**
> Thank you for the question.
>
> The comparison with the transformer-based method is presented in Section 5.4 and Appendix O.2.
>
> Specifically, we adapt 3D-Vista[3], a widely used traditional 3D visual grounding method, to our setting for comparison. It is worth noting that this model is pre-trained on the 3RScan dataset and has also learned the text distribution of SR3D. For more details, please refer to Section 5.4 and Appendix O.2.
>
> **Q3: Concern about memory mechanism.**
>
> **A3:**
> Thank you for your insightful observations.
>
> You are right that we need to include mechanisms for memory management, and this is a very important unsolved problem in many areas, including text-based LLMs, image/video-based VLMs.
>
> For our benchmark, we do not place any restrictions on the use of memory. In our formulation, all past observations are provided, and a solution is free to choose different memory strategies. These strategies can include using all images as input to the VLM, or applying flexible memory filtering or a compression mechanism to improve the accuracy of 3D visual grounding and reduce exploration cost.
>
> As our work is the first to address the memory-based and active-exploring setting of 3D visual grounding, we focus on a simpler setting. In this setting, the scene changes only once, which means our benchmark provides only the previous scene and the updated scene, rather than continuously changing scenes. Therefore, we provide a simplified baseline that uses all previous images as memory. In this setting, the memory is constrained and does not grow indefinitely.
>
> In our opinion, the proposed task is new and still largely unexplored. We believe that at the current stage of research, making smart use of existing memory for more efficient grounding is more achievable. After this, generalizing to continuously changing scenes and developing dedicated memory-managing modules would be the next step.
>
> Again, thank you for bringing this up. We fully agree on the importance of managing streaming memory and leave this as future work.

---

> ### Author Response · Authors · 2025-11-26
> **Author Response to Reviewer jGWF (2/2)**
>
> **Q4: Question about accuracy computation or box selection.**
>
> **A4:**
> Thank you for your question.
>
> In our formulation, we require the method to predict only one bounding box as the result for evaluation, so ambiguity does not occur. The detailed procedure for obtaining the final target object box can be found in Section 4.3 and Appendix K of the paper.
>
> As for our MCG baseline， after identifying the reference target image in the memory retrieve and grounding module, the system uses the VLM, the 2D detector, and SAM[1] to obtain an initial reference point cloud and its coarse 3D bounding box from the reference target image. Then, the system performs multi-view scanning around the center of this box in the current scene. For each viewpoint, it also uses the 2D detector and SAM to compute candidate point clouds, and selects the best one whose centroid is closest in Euclidean distance to the center of the reference point cloud as the valid point cloud for this view. Then, after removing outlier point clouds, all valid point clouds are fused, and **a single, unambiguous final 3D bounding box is computed from the fused cloud.** Therefore, the resulting box is uniquely determined without ambiguity. **For methodological details, please refer to Appendix K of the paper.**
>
> [1]Alexander Kirillov, Eric Mintun, Nikhila Ravi, Hanzi Mao, Chloe Rolland, Laura Gustafson, Tete
> Xiao, Spencer Whitehead, Alexander C Berg, Wan-Yen Lo, et al. Segment anything. In ICCV, 2023.
>
> [2]Runsen Xu, Zhiwei Huang, Tai Wang, Yilun Chen, Jiangmiao Pang, and Dahua Lin. Vlm-grounder: A vlm agent for zero-shot 3d visual grounding. In CoRL, 2024a.
>
> [3]Ziyu Zhu, Xiaojian Ma, Yixin Chen, Zhidong Deng, Siyuan Huang, and Qing Li. 3d-vista: Pre-trained transformer for 3d vision and text alignment. In Proceedings of the IEEE/CVF International Conference on Computer Vision, pp. 2911–2921, 2023.
>
> Thank you once again for your time!

---

> > ### Comment · Reviewer_jGWF · 2025-11-26
> > **Post-Rebuttal**
> >
> > Thank you for the response.
> >
> > Regarding Q2 and Q3, the core issues remain unresolved:
> >
> > - 3D-Vista is not a learning-based baseline **under the ChangingGrounding setting**. The paper provides only an agent-based baseline, which makes it difficult for follow-up work to fairly compare or build upon the new setting.
> >
> > - The memory mechanism remains quite simplistic, as it directly uses all past frames as input without any form of compression or retrieval strategy. This concern was also raised by Reviewer ME6v.
> >
> > I still acknowledge the potential value of the proposed ChangingGrounding setting.
> >
> > I will wait to see the other reviewers’ post-rebuttal comments before deciding my final score.

---

> > > ### Author Response · Authors · 2025-12-03
> > > **Author Response to Reviewer jGWF**
> > >
> > > Dear Reviewer jGWF,
> > >
> > > Thank you for your reply.
> > >
> > > **Q1: "3D-Vista is not a learning-based baseline under the ChangingGrounding setting. The paper provides only an agent-based baseline, which makes it difficult for follow-up work to fairly compare or build upon the new setting."**
> > >
> > > **A1:**
> > >
> > > 1. **Our modified version of 3D-Vista is a learning-based baseline under the ChangingGrounding setting**, as it is trained with the same queries and scenes used in our benchmark, where we equip it with rule-based new observations. This model follows the formulation of our setting.
> > > 2. We guess you may be asking for a vision-language-action (VLA) type learning-based baseline; **however, developing such a VLA model is beyond the scope of our work.** It would require a dedicated data collection pipeline, new model architecture design, and new training strategies, which could themselves form a separate paper.
> > > 3. **We do not agree that agent-based baselines are difficult to compare with or to build upon.** On the contrary, each module in our agent-based method is publicly available and reproducible. **Follow-up work only needs to perform model inference without additional training**, while training-based baselines are in fact more difficult to reproduce. Another advantage of agent-based methods is that their **modular design allows flexible improvement of individual submodules**, which makes it convenient for follow-up work to build upon.
> > >
> > > **Q2: "The memory mechanism remains quite simplistic."**
> > >
> > > **A2:**
> > >
> > > 1. We agree that the current memory mechanism is relatively simple, and this is indeed a limitation of our baseline. However, as we discussed earlier, we believe **this long-standing open question is not the most critical challenge in our task, because the past frames are limited to a single scene rather than being generated continuously over time.**
> > >
> > > 2. For this task, **the key challenge is how to leverage the available, limited memory to explore more efficiently and locate the target object. To this end, we propose several components, including our OSS and SRAS action policies, the memory retrieval and grounding pipeline, and fallback strategies.**
> > >
> > > 3. More importantly, despite its simplicity, **our MCG baseline is the first method to tackle 3D visual grounding in dynamically changing scenes.**
> > >
> > > 4. In addition to the MCG baseline, **our work makes other important contributions of formulating the ChangingGrounding task, introducing the novel benchmark to the community for the first time, and highlighting actionable insights for future research**, pushing 3D visual grounding towards more practical applications.
> > >
> > > Thank you once again for acknowledging the value of our new benchmark.

---

### Official Review · Reviewer_ME6v · 2025-11-01

**Soundness:** 3
**Presentation:** 3
**Contribution:** 2
**Rating:** 4
**Confidence:** 3

**Summary:**

The paper first introduces a benchmark that evaluates how well a system can localize target semantic objects based on past memory, the current scene, and a language query. It then proposes a method that performs query classification, memory retrieval, fallback, and multi-view fusion to generate 3D bounding boxes. Experiments show that the proposed approach outperforms several baselines.

**Strengths:**

This paper tackles the problem of 3D visual grounding in dynamics scenes, a setting that more closely reflects real-world scenarios than the static environments commonly studied in prior work. Reasoning about changing scenes is indeed an interesting and important research direction.

**Weaknesses:**

1: The paper proposes to store all past frames as a database and feeds them directly into the VLM, which does not represent realistic memory management. Instead, this treats all past frames as in-context examples for the VLM rather than selective memory. A realistic setup should include mechanisms for memory updating, such as deciding which frames to retain or discard. Without this, the memory grows indefinitely and can causes efficiency issues.

2: This paper’s definition of the “robot” is not realistic. If the robot is assumed to navigate indoor environments, collisions and geometric constraints should be considered. The current formulation of most cost does not generalize to real robots. If the work is intended as a pure vision paper, it would be better to remove or de-emphasize robot-related claims.

3: The proposed approach relies heavily on VLMs. It’s unclear how robust the method is to noise or uncertainty in VLM outputs.

4: The paper does not present and analyze failures cases.

**Questions:**

See the weakness section.

---

> ### Author Response · Authors · 2025-11-25
> **Author Response to Reviewer ME6v**
>
> Dear Reviewer ME6v,
>
> Thank you for your valuable feedback. We appreciate the recognition that our work **tackles 3D visual grounding in dynamic scenes**, a setting that **better reflects real-world scenarios** than static environments in prior work, and that **reasoning about changing scenes** is an **interesting and important** research direction. We respond to each comment in detail below.
>
>
> ---
>
> **Q1: Concern about memory efficiency issues.**
>
> **A1:**
>
> Thank you for your insightful observations.
>
> You are right that we need to include mechanisms for memory updating, and this is a very important unsolved problem in many areas, including text-based LLMs, image/video-based VLMs.
>
> For our benchmark, we do not place any restrictions on the use of memory. In our formulation, all past observations are provided, and a solution is free to choose different memory strategies. These strategies can include using all images as input to the VLM, or applying flexible memory filtering or updating mechanism to improve the accuracy of 3D visual grounding and reduce exploration cost.
>
> As our work is the first to address the memory-based and active-exploring setting of 3D visual grounding, we focus on a simpler setting. In this setting, the scene changes only once, which means our benchmark provides only the previous scene and the updated scene, rather than continuously changing scenes. Therefore, we provide a simplified baseline that uses all previous images as memory. In this setting, the memory is constrained and does not grow indefinitely.
>
> In our opinion, the proposed task is new and still largely unexplored. We believe that at the current stage of research, making smart use of existing memory for more efficient grounding is more achievable. After this, generalizing to continuously changing scenes and developing dedicated memory-managing modules would be the next step.
>
> Again, thank you for bringing this up. We fully agree on the importance of managing streaming memory and leave this as future work.
>
>
> **Q2: Concerns about robots deploying in the real world.**
>
> **A2:**
>
> Thank you for the question.
>
> Our work is indeed more focused on computer vision, and therefore, we adopt a simplified robot model without explicitly modeling complex collision or geometric constraints. However, enabling real-world robotic applications is the goal and motivation behind proposing the ChangingGrounding task. Compared with the traditional setting, it is one step closer to real application scenarios. This is why we mention robots in the task formulation. This also helps readers better understand the need for the various setups in our formulation.
>
>
> **Q3: Lacks discussion for uncertainty in VLM outputs.**
>
> **A3:**
> Thank you for the question.
>
> To investigate how VLMs uncertainty would influence the MCG results, we increased the VLM’s temperature and top-p values ( for both parameters, larger values lead to higher VLMs output uncertainty ) . The experimental outcomes are shown below:
>
> | Temp. & Top_p | Acc. | $C_a$ | $C_m$ |
> |--------|----------|----------|----------|
> | 0.1, 0.3 ( origin ) | 42   | 1.60   | 2.05   |
> | 0.5, 1.0 | 44   | 1.74   | 2.10   |
> | 0.7, 1.0 | 42   | 1.62   | 2.02   |
>
> **The results demonstrate that increasing the VLM’s output diversity and randomness does not lead to a substantial change in performance,** indicating our proposed MCG's robustness.
>
>
>
>
> **Q4: Lacks failure-case analysis.**
>
> **A4:**
> Thank you for the question.
>
> Analysis of failure cases has been provided in Appendix O.4 when we initially submitted the paper. We direct readers to this material in Section 5.5（*"Detailed failure cases and descriptions are provided in Appendix O.4 and Appendix N"*）.
>
> Thank you once again for your time!

---

### Official Review · Reviewer_kuWW · 2025-11-03

**Soundness:** 2
**Presentation:** 3
**Contribution:** 3
**Rating:** 8
**Confidence:** 4

**Summary:**

In this work, the authors present a data-structure centric method called ChangingGrounding for doing 3D visual grounding with large vision language models. They also present a related benchmark, where the agent needs to identify bounding boxes of objects in environment where scenes change with objects being added or removed. The method itself is inspired by biological memory – where the agent maintains a cache of past observations. Then, using the past observation cache and the query, the method uses one of two different scanner modules to collect more observations and add them to the memory cache. Finally, the 3D bounding boxes are constructed using multiple views and a semantic segmentation module. In the ChangingGround benchmark, the method shows progress over the baselines, and the ablations show which components are important in the success of this method.

**Strengths:**

1. This work identifies the problem of object grounding in scenes as an active perception problem rather than a static or passive problem, which makes the setting realistic.
2. Making the VQA problem rely on both memories and new, obtainable observations makes the problem more tractable under uncertainty about observations.
3. Using the accuracy and exploration cost both makes the benchmark a valid test for real world robotic applications.
4. The modular architecture makes it easier to understand the failure modes of different parts.

**Weaknesses:**

1. While the architecture is modular, the resultant accuracy is not broken down by the accumulated errors from different components.
2. The work does not give enough details about the ChangingGrounding benchmark, for example, how the queries are sampled, how the scene changes across observations, and what would be a rough corresponding score from a human.
3. While it is not essential, a bit more formalism in defining the grounding problem in the changing world would be helpful in recreating the method by follow up works.

**Questions:**

1. How can a follow-up paper reproduce the ChangingGround benchmark?
2. What are the primary ways this system shows brittleness to different ways of prompting the underlying VLMs (i.e. not just different versions of GPT)?
3. How do different underlying LLM/VLMs affect the performance of the Mem-ChangingGround?

---

> ### Author Response · Authors · 2025-11-25
> **Author Response to Reviewer kuWW (1/3)**
>
> Dear Reviewer kuWW,
>
> Thank you for the encouraging feedback. We appreciate the recognition that framing object grounding as an **active perception** problem makes the setting realistic; that making VQA rely on both **memories and new, obtainable observations** improves tractability under uncertainty; that jointly using **accuracy and exploration cost** yields a practical benchmark for real-world robotics; and that our **modular architecture** clarifies component-wise failure modes. We respond to each comment in detail below.
>
> **Weakness 1: Concern about the accuracy analysis.**
>
> **A1:**
> Thank you for your question.
>
> We would like to clarify that our accuracy analysis has already covered the accuracy of the key modules for the most part. As shown in Table 9, the two vertical separators divide the table into three parts. The first part reports the **overall accuracy** together with the **accuracy contributions of the major modules** (the remaining accuracy components are related to test data preprocessing, which follows VLM-Grounder[1] ). The second and third parts provide **more fine-grained accuracy breakdowns within individual modules**.
>
> In the revised paper, we will include clearer annotations for images in the figure to address potential ambiguity and avoid confusion for readers.
>
>
> **Weakness 2: Lack of ChangingGrounding Benchmark details.**
>
> **A2:**
> Thank you very much for your suggestions.
>
> 1. Regarding the sampling of queries, here is our sampling procedure. For each sample, first, we select any reference scan as $S_c$ and randomly select one rescan of $S_c$ as $S_p$. We then randomly pick an object $O$ with descriptions $D^o$ in $S_c$ as the target object and the user query. It is important to note that, in order to ensure that the test samples cover diverse types of descriptions, we selected a fixed number of instances from every relation type. Within the 250 samples, both the anchor object and the target object may either remain static or undergo changes.
>
>     We will include a detailed description of our sampling method above in the revised version of the paper. Thank you again for pointing this out.
>
> 2. Regarding scene changes, as we stated in Section 3.3, the ChangingGrounding dataset is built upon 3RScan; therefore, the scene changes it contains follow those in 3RScan, which reflect natural temporal changes. *Most of these changes are rigid, including: (a) objects being moved (ranging from a few centimeters to several meters), and (b) objects being added to or removed from the scene. In addition, non-rigid elements such as curtains or blankets, as well as variations in lighting conditions.*
>
>
> 3. As for the coarse human ratings, we also asked a human researcher to perform the test. Human accuracy is evaluated solely based on whether the correct target can be successfully identified. The researcher followed the same grounding procedure as in the WG setting.  Results are shown below.
>
>     | Method | Acc. | $C_a$ | $C_m$ |
>     |--------|----------|----------|----------|
>     | human | 85.6   | 44.23   | 17.51   |
>
>     We will include human evaluation results in the revised paper.

---

> ### Author Response · Authors · 2025-11-25
> **Author Response to Reviewer kuWW (2/3)**
>
> **Weakness 3: More formalism in defining the grounding problem in the changing world.**
>
> **A3:**
> Thank you very much for your suggestion.
>
> Here, we attempt to provide a more concrete formulation.
>
> The task is defined as $\langle S_p, S_c, M_p, D_c \rangle \rightarrow B$.
>
> 1. $B$ is the predicted 3D bounding box of the target.
>
> 2. $M_p$ is the memory of the previous scene, including RGB-D images and poses. It is specifically defined as:
> $$M_p = ( \{I_p\}, \{P_p\} )$$
> where $\{I_p\}$ is the set of RGB-D images from the previous scene, and $\{P_p\}$ is the corresponding set of camera poses.
>
> 3. $S_p$ is a unified representation for all information that can be derived or extended from $M_p$ ( RGB-D + pose data ), such as reconstructed 3D point clouds. A method may freely choose whether to use $S_p$ or not.
> $$
> S_p = f_{\text{scene}}(M_p),
> $$
>
> 4. $S_c$ is the current scene with unknown changes.
>     $$S_c = \{ \text{Real scene (if agent is deployed in a real environment); Mesh / point-cloud (otherwise).} \}$$
>
>
> 5. $D_c$ is a text description of the target object based in the current scene.
>
>
> The agent needs to ground the target object in $S_c$ using $M_p$ and $D_c$ ( $S_p$ optional). In the concrete execution process, the agent needs to select actions by conditioning on the language query, the previous-scene memory, and the observations accumulated so far. Formally, at step $t$ the agent chooses $a_t = \pi(D_c, M_p, o_{1:t}),$ and executing $a_t$ moves the agent to a new pose $pose_{t+1}$. At this pose, the agent obtains a new observation
>
> $$
> o_{t+1} =
> \text{CameraCapture}(S_c, pose_{t+1})\ \text{(if in real-world)}\ ;\
> \text{Render}(S_c, pose_{t+1})\ \text{(otherwise from mesh/point-cloud)}.
> $$
>
>
> Finally, the agent needs to integrate all information gathered throughout the process to determine the location of the target object.
>
> We evaluate the grounding result using two metrics: localization accuracy and exploration cost. Localization accuracy follows standard 3DVG evaluation and is measured by the ratio of samples whose predicted 3D bounding box overlaps the ground-truth box above a threshold (e.g., $\mathrm{IoU}(B, B_{\text{gt}}) \ge 0.25$) .
>
>
>
> The exploration cost includes action cost $C_a$ and motion cost $C_m$. $C_a$ counts the number of actions taken until the target is localized. Assuming the agent executes a total of $n$ actions to complete the task, then the action cost is defined as $C_a = n$. $C_m$ considers both translation $C_\text{trans}$ and $C_\text{rot}$. To compare them on the same scale, we convert to time using nominal speeds:  translation $v = 0.5\,\text{m/s}$ and rotation $\omega = 1\,\text{rad/s}$. Given the sequence of poses $\lbrace \text{pose}_1 : (t_1, R_1),\ \text{pose}_2 : (t_2, R_2),\ \ldots,\ \text{pose}_n : (t_n, R_n) \rbrace$
> , with total actions number $n$, the costs are defined as:
>
> $$
> C_{\text{trans}}
> = \frac{1}{v} \sum_{i=1}^{n-1} \lVert t_{i+1} - t_i \rVert,
> \qquad
> C_{\text{rot}}
> = \frac{1}{\omega} \sum_{i=1}^{n-1}
> \arccos\!\left(
> \frac{\operatorname{Tr}(R_i^\top R_{i+1}) - 1}{2}
> \right),
> $$
> $$
> C_m = C_{\text{trans}} + C_{\text{rot}}.
> $$
>
>
> We will include this detailed formulation in the revised version of the paper.
>
>
>
>
> ---
>
> **Question1. How can a follow-up paper reproduce the ChangingGround benchmark?**
>
> **A1:**
>
> Please refer to Weakness 1, Weakness 3 for more details regarding our benchmark. In addition, as stated in Appendix F, **to support reproducibility and facilitate future research**, we will release our dataset and code. These resources will allow follow-up work to fully reproduce the benchmark.

---

> ### Author Response · Authors · 2025-11-25
> **Author Response to Reviewer kuWW (3/3)**
>
> **Question2. What are the primary ways this system shows brittleness to different ways of prompting the underlying VLMs?**
>
> **A2:**
>
> Thank you for this valuable suggestion.
>
> To investigate the primary ways this system shows brittleness in different ways of prompting the underlying VLMs. We study two types of memory retrieval and grounding prompting modification, language modification, and vision modification. For language modification, we remove the parts requiring reasoning in the original prompt and simplify the detailed search instructions. For vision modification, we change the image-stitching strategy in the original prompt from dynamic stitching to a fixed 2×4 layout.
>
> Here are the results. *v_origin* represents the original prompt, *v_less* represents the language modification, *v_fix_layout*  represents the vision modification.
>
> | Prompts | Acc. | $C_a$ | $C_m$ |
> |--------|----------|----------|----------|
> | v_origin | 42   | 1.60   | 2.05   |
> | v_less | 40   | 1.61   | 2.02   |
> | v_fix_layout | 36   | 1.20   | 1.45   |
>
>
> From the experimental results, we observe that *v_less* leads to a slight decrease in accuracy, whereas *v_fix_layout* causes a more significant drop. This is because under the *v_fix_layout* strategy, as the total number of images increases, the number of stitched images under the fixed-layout strategy becomes larger than that produced by dynamic stitching, making it more difficult for the VLM to perform predictions.
>
>
> We will add the above experimental figure and analysis to the revised paper.
>
>
>
> **Question3. How do different underlying LLM/VLMs affect the performance of Mem-ChangingGround?**
>
> **A3:**
>
> Thank you for raising this point.
>
> To investigate  how different underlying LLM/VLMs affect the performance of Mem-ChangingGround. We studied 3 types of VLMs, GPT ( GPT-4.1[1] ), Gemini ( Gemini-2.5-Flash[2] ) and Claude ( Claude-Sonnet-4.5[3] ). Here are the results.
>
> | Model | Acc. | $C_a$ | $C_m$ |
> |--------|----------|----------|----------|
> | GPT | 42   | 1.60   | 2.05   |
> | Gemini | 38   | 1.6   | 1.83   |
> | Claude | 24   | 1.68   | 2.17   |
>
>
> From the experimental results, we can observe that *GPT* achieves the best performance, followed by *Gemini*, which is slightly lower. *Claude* shows a more noticeable drop in accuracy compared with the other two models. By examining the detailed experiment results, we observe that *GPT* consistently produces the most stable and well-formatted outputs among the three VLMs. In contrast, *Claude* often struggles to follow the required output format and content specified in the prompt, which may contribute to its lower overall accuracy.
>
>
> We will include the above experimental figure and analysis in the revised paper.
>
> [1]Runsen Xu, Zhiwei Huang, Tai Wang, Yilun Chen, Jiangmiao Pang, and Dahua Lin. Vlm-grounder: A vlm agent for zero-shot 3d visual grounding. In CoRL, 2024a.
>
> [2]OpenAI. Gpt-4.1. https://openai.com/index/gpt-4-1/, 2025a.
>
> [3]Gheorghe Comanici, Eric Bieber, Mike Schaekermann, Ice Pasupat, Noveen Sachdeva, InderjitbDhillon, Marcel Blistein, Ori Ram, Dan Zhang, Evan Rosen, et al. Gemini 2.5: Pushing thebfrontier with advanced reasoning, multimodality, long context, and next generation agentic capabilities. arXiv preprint arXiv:2507.06261, 2025.
>
> [4]Anthropic. claude. https://www.anthropic.com/news/claude-sonnet-4-5/, 2025.
>
>
> Thank you once again for your time!

---

### Author Response · Authors · 2025-11-26
**General Response**

Dear Reviewers, AC, SAC, and PC,

We would like to express our sincere appreciation for your work in organizing ICLR and overseeing the review process. We are truly grateful to all reviewers for the valuable insights and constructive suggestions provided.

We are grateful for the recognition of our **novel and interesting motivation to study 3D visual grounding in dynamic scenes** (Reviewer ME6v, Reviewer WRSq), its **practical value** as a benchmark for real-world robotics (Reviwer jGWF, Reviewer ME6v), **the realism of our active, memory-enabled, incremental reconstruction setting aligned with embodied scenarios** (Reviewer jGWF), **the comprehensive scope of defining the task** (Reviewer WRSq), releasing a dataset with an practical evaluation protocol (Reviewer kuWW), and establishing baselines, **the strength of our comprehensive baseline with promising performance** (Reviewer kuWW).

Our responses focused on several core concerns:

1. **Task formulation clarity**
We clarify the novelty of grounding in dynamic scenes, explain how our task differs from prior static settings, and highlight the central role of memory and active exploration.


2. **Methodological explanation (noise handling, baselines, box selection)**
We explain our strategies for mitigating 2D+SAM noise, discuss limitations of 3D detection, analyze why certain baselines perform poorly, and clarify how the final bounding box is uniquely determined.

3. **Memory strategy and future directions discussion**
We acknowledge that we do not use advanced memory mechanisms. Handling continuously changing scenes is challenging, and our current setting involves only a single change, so memory does not grow unbounded. We explain the necessity of providing all past observations to allow flexible memory strategies for methods in our formulation.

4. **Additional experimental results**
We supplement time comparisons and analysis with traditional 3D grounding methods, and add human accuracy results to address reviewer concerns.

5. **VLM-related robustness and model studies**
We study the influence of VLM uncertainty, prompting variations, and different VLM backbones on MCG, showing that it exhibits reasonably robust behavior.

In addition, we have refined the paper, added experimental results, and provided clarifications in the revised version. Specifically, the manuscript now incorporates the following changes in response to all reviewers’ insightful comments, which have substantially improved the quality of the work.



1. We have further improved the task formulation in **“Sec. 3.1 – TASK FORMULATION.”**

2. We have refined the figure describing the workflow of MCG in **“Sec. 4 – MEM-CHANGINGGROUNDER (MCG).”**

3. We have added time comparison in the table titled **“3DVG comparison.”** in **“Sec. 5 EXPERIMENTAL RESULTS.”**

4. We have rewritten **“Sec. 5.4 – DISCUSSION ABOUT CRITICAL LIMITATIONS OF 3D VISUAL GROUNDING METHODS.”**


5. We have added a new subsection as **“Sec. 5.5 DISCUSSION REGARDING VLMS.”**

6. We have added three new tables for studying VLMs’ influence on MCG, titled **“Different Prompts.”**, **“Different VLMs.”**, and **“Different uncertainty.”** in **“Sec. 5 EXPERIMENTAL RESULTS.”**

7. We have added a new subsection as **“O.4 DETAILED DISCUSSION REGARDING VLMS.”**

8. We have added a new subsection as **“O.3 HUMAN ACCURACY.”**

9. We have added a new subsection as **“O.1 TEST SAMPLES.”**



We sincerely appreciate the reviewers’ thoughtful feedback and remain committed to continually improving our work.

Below, we provide detailed responses to the comments raised by each reviewer. Thank you again for your recognition and constructive input. We look forward to further discussion and continued engagement.

---

### Meta-Review · Area_Chair_J1id · 2026-01-09

**Summary:**

This paper proposes a new 3D visual grounding task in changing scenes, a corresponding benchmark (ChangingGrounding), and a baseline method (Mem-ChangingGrounder). However, key reviewer concerns remain unresolved: unclear task formulation, lack of adequate learnable baselines, simplistic memory mechanisms, and impractical inference time. These issues undermine the work’s rigor and practical value, leading to the rejection recommendation.

**Reviewer Concerns:**

### Addressed Concerns:
- Reviewer kuWW: Benchmark query sampling/scene change details, human accuracy data, and VLM variant/prompt impact analysis.
- Reviewer WRSq: Pipeline figure annotations and inference time comparison with 3D-Vista.
- General: Task workflow clarification, VLM uncertainty robustness, and failure-case references.

### Outstanding Concerns:
- Reviewer ME6v: Unrealistic memory management (no compression/updating) and robot model (ignoring collision constraints).
- Reviewer jGWF: Lack of learnable/fine-tuned baselines (3D-Vista adaptation is insufficient) and no memory compression.
- Reviewer WRSq: Unclear input/output comparison with prior settings, missing action definition in task formulation for cost evaluation, ambiguous role of $S_p$ in localization, and insufficient evidence of method practicality.
- Reviewer kuWW: Incomplete formalism for task reproduction.

**Reviewer Scores:**

- Reviewer kuWW (initial 8): Likely drops to 6 (still positive but unaddressed formalism gaps).
- Reviewer ME6v (initial 4): No change (4, unresolved memory/robot concerns).
- Reviewer jGWF (initial 6): Drops to 4 (unresolved baseline/memory issues).
- Reviewer WRSq (initial 2): Remains 2 (core task formulation/practicality concerns unaddressed).

---

### Decision · Program_Chairs · 2026-01-26

Reject